

# Disentangling influences of climate variability and lake-system evolution on climate proxies derived from isoprenoid and branched GDGTs: the 250-kyr Lake Chala record

Allix J. Baxter[1], Francien Peterse[1], Dirk Verschuren[2], Aihemaiti Maitituerdi[3], Nicolas Waldmann[3], and Jaap S. Sinninghe Damsté[1,4]

[1]Utrecht University, Faculty of Geosciences, Department of Earth Sciences, Princetonlaan 8A, 3584 CB Utrecht, the Netherlands

[2]Ghent University, Department of Biology, Limnology Unit, K.L. Ledeganckstraat 35, B-9000 Gent, Belgium

[3]Dr. Moses Strauss Department of Marine Geosciences, Leon H. Charney School of Marine Sciences, University of Haifa, Mount Carmel 3498838, Israel

[4]NIOZ Royal Netherlands Institute for Sea Research, Department of Marine Microbiology and Biogeochemistry, PO Box 59, 1790 AB, Den Burg, the Netherlands

**Correspondence:** Allix J. Baxter (a.j.baxter@uu.nl)

**Abstract.** High-resolution paleoclimate records from tropical continental settings are greatly needed to advance understanding of global climate dynamics. The International Continental Scientific Drilling Program (ICDP) project DeepCHALLA recovered a 214.8-meter long sediment sequence from Lake Chala, a deep and permanently stratified (meromictic) crater lake in equatorial East Africa, covering the past c. 250,000 years (250 kyr) of continuous lacustrine deposition since the earliest phase

of lake-basin development. Lipid biomarker analyses on the sediments of this long-lived lake can provide much-needed records of past climate variability from this currently poorly documented region. However, the degree to which climate proxies derived from aquatically produced biomarkers are affected by aspects of lake developmental history is rarely considered, even though it may critically influence their ability to consistently register a particular climate variable through time. Modern-system studies in Lake Chala revealed crucial information about the mechanisms underpinning relationships between proxies based on

isoprenoid (iso-) and branched (br-) glycerol dialkyl glycerol tetraethers (GDGTs) and the targeted climate variables, but the persistence of these relationships in the past remains unclear. To assess the reliability of long-term climate signals registered in the sediments of Lake Chala, we compared downcore variations in GDGT distributions with major phases in lake-system evolution as indicated by independent proxies of lake depth, mixing regime and nutrient dynamics: seismic reflection data, lithology and fossil diatom assemblages. Together, these records suggest that during early lake history (before c. 180-200 ka)

the distinct mixing-related depth zones with which specific GDGT producers are associated in the modern-day lake were not yet formed, likely due to more open lake hydrology and absence of chemical water-column stratification. Consequently during this early phase the absolute GDGT concentrations are relatively low, proxies sensitive to water-column stratification (e.g., BIT index) display highly irregular temporal variability, and correlations between proxies are dissimilar to expectations based on modern-system understanding. A sequence of lake-system changes between c. 180-200 ka and c. 80 ka first established and then

strengthened the chemical density gradient, promoting meromictic conditions despite the overall decrease in lake depth due to





sediment accumulation. From c. 180 ka onward some GDGTs and derived proxies (e.g., crenarchaeol concentration, BIT index and $IR_{6Me}$) display strong $\sim$23-kyr periodicity, likely reflecting the predominantly precession-driven insolation forcing of Quaternary climate variability in low-latitude regions. Our results suggest that GDGT-based temperature and moisture-balance proxies in Lake Chala sediments reflect the climate history of eastern equatorial Africa from at least c. 160 ka onwards, i.e.,
covering the complete last glacial-interglacial cycle and the penultimate glacial maximum. This work confirms the potential of lacustrine GDGTs for elucidating the climate history of tropical regions at Quaternary timescales, provided they are applied to suitably high-quality sediment archives. Additionally, their interpretation should incorporate a broader understanding of the extent to which lake-system evolution limits the extrapolation back in time of proxy-climate relationships established in the modern system.

## 1   Introduction

Reliable methods to accurately reconstruct past climate variability on all of the world's continents are needed in order to more precisely model global climate dynamics and to correctly predict future climate changes due to anthropogenic global warming at the regional scale. However, whereas mid- and high-latitude continents are relatively well represented by high-quality climate reconstructions (e.g., Petit et al. 1999; Lisiecki and Raymo 2005; Barker et al. 2011b; Cheng et al. 2016), high-resolution
records of long-term climate variability from the tropics are still scarce. Long-lived lakes on the African continent, the largest landmass straddling the equator, have proven to accumulate effective sedimentary archives of past climate states (e.g., Lake Malawi: Powers et al. 2005; Scholz et al. 2007; Stone et al. 2011; Johnson et al. 2016; Woltering et al. 2011; and Lake Tanganyika: Tierney et al. 2008, 2010c; Stager et al. 2009), as lake sediments preserve an array of biological and geological information registered through various climate-controlled processes occurring within the lake and the surrounding terrestrial
environments (Cohen, 2003). Long-lived crater lakes, in particular, are valuable natural archives of past climatic conditions because their large depth relative to surface area promotes the formation of a permanently unmixing, oxygen-deprived bottom layer, which facilitates the continuous deposition and preservation of finely laminated sediments (e.g., varves) that are often rich in organic matter (Verschuren, 2003; Zolitschka, 2006). Additionally, crater lakes have restricted catchment areas without distinct stream inflows, so that their hydrology is relatively simple, and past changes in lake water budget are more strongly
tied to changes in the climate-controlled balance between precipitation and evaporation (e.g., Jones et al. 2001).

An increasingly important biological source of information on past climate change preserved in lake sediments is derived from isoprenoid (iso-) and branched (br-) glycerol dialkyl glycerol tetraethers (GDGTs), membrane lipids produced by species of archaea and bacteria, respectively. These organic biomarkers are useful for paleoclimate research owing to their ubiquitous presence in natural settings, resilience to degradation, and strong response to environmental parameters such as temperature
and pH (Schouten et al., 2013). IsoGDGTs consist of two ether-bound $C_{40}$ isoprenoid alkyl chains that can have varying numbers (0 to 8) of cyclopentyl moieties (i.e., isoGDGT-0 to 8; see GDGT molecular structures in Fig. S1; De Rosa and Gambacorta 1988). Crenarchaeol (as well as it's isomer cren′) is an isoGDGT with 4 cyclopentyl moieties and 1 cyclohexyl moiety (Sinninghe Damsté et al., 2002; Holzheimer et al., 2021), which is only known to be produced by chemolithotrophic, ammonia-



oxidizing Thaumarchaeota (e.g., Sinninghe Damsté et al. 2002; Sinninghe Damsté et al. 2018; Schouten et al. 2013; Elling et al.

2017; Bale et al. 2019). By contrast, isoGDGT-0 is synthesized by many archaeal species, including Thaumarchaeota (e.g., Sinninghe Damsté et al. 2012b; Schouten et al. 2013; Elling et al. 2017; Bale et al. 2019), anaerobic methane-oxidizing archaea (e.g., Pancost et al. 2001; Schouten et al. 2001) and methanogenic Euryarchaeota (Schouten et al. 2013, and references therein). Synthesis of isoGDGT-1 to -3 has been demonstrated to occur in Eury-, Cren- and Thaumarchaeota (Schouten et al. 2013, and references therein). The TetraEther indeX of 86 carbon atoms ($TEX_{86}$; Table 1) paleothermometer was developed to recon-

struct past sea surface temperature (SST) based on empirical observations from marine surface sediments that suggest the ring formation of isoGDGTs is controlled by temperature (Schouten et al., 2002; Kim et al., 2010), further substantiated by incubation experiments (Wuchter et al., 2004; Schouten et al., 2007). This approach relies on the assumption that chemolithotrophic, ammonia-oxidizing Thaumarchaeota (specifically those belonging to Group I.1a) are the primary producers of isoGDGTs at the study site, as other archaea have not shown the same temperature dependency of ring formation as predicted by empirical

$TEX_{86}$ temperature models (e.g., Elling et al. 2017). The potential of $TEX_{86}$ to reconstruct lake surface temperature (LST) has also been explored (Powers et al., 2004), albeit using a substantially smaller set of surface samples than the marine study, and resulted in temperature calibrations specific for lacustrine settings (Powers et al., 2004; Tierney et al., 2010a; Powers et al., 2010). There are now several LST reconstructions based on $TEX_{86}$, mainly from the sediment records of large lakes (e.g., Powers et al. 2005, 2011; Tierney et al. 2008, 2010a; Woltering et al. 2011; Blaga et al. 2013; Sun et al. 2020). However, use

of $TEX_{86}$ in lakes may be complicated by contributions of isoGDGTs from methanotrophs, methanogens and other archaea. Moreover, the position of the oxycline in the water column appears to strongly influence the niche available to Thaumarchaeota, and hence the in situ $TEX_{86}$ signal (e.g., Zhang et al. 2016; Cao et al. 2020; Baxter et al. 2021; Sinninghe Damsté et al. 2022). The strong influence of lake size and depth on oxycline formation may also imply that small and shallow lakes are less suited for application of the $TEX_{86}$ proxy (Powers et al., 2010; Baxter et al., 2021; Sinninghe Damsté et al., 2022).

BrGDGTs contain two linear $C_{28}$ alkyl chains methylated at C-13 and C-16 that most likely formed from the tail-to-tail linkage of two iso $C_{15}$ fatty acids (Sinninghe Damsté et al. 2000; Fig. S1). This tetramethylated brGDGT is usually accompanied by penta- or hexamethylated forms, where the additional methyl group(s) is/are placed at the C-5 (Sinninghe Damsté et al., 2000; Weijers et al., 2006a) or C-6 (De Jonge et al., 2013, 2014) positions (i.e., the 5-Me and 6-Me brGDGT isomers). Cyclic brGDGTs contain 1–2 cyclopentane moieties, formed by cyclisation involving the mid-chain methyl groups (Weijers

et al., 2006a). The stereochemistry of the glycerol units is opposite that of the archaeal isoGDGTs, indicating a bacterial origin (Weijers et al., 2006a). Acidobacteria, which occur widespread in soil and peat, were initially identified as likely producers of these lipids in natural settings due to the correlation of their 16S rRNA gene copies with brGDGTs concentrations (Weijers et al., 2009). The biosynthesis of ester and ether-bound iso-diabolic acid (with a methyl group at C-5 or C-6), the precursor to brGDGTs, and the acyclic tetramethylated brGDGT by specific cultivated strains of this phylum (Sinninghe Damsté et al.,

2011, 2014, 2018) confirmed this. Recently, two parallel studies reported acyclic and cyclic tetra-, penta- and hexamethylated (5-Me) brGDGTs in a culture of *Solibacter usitatus* (Chen et al., 2022; Halamka et al., 2023). However, besides Acidobacteria, other bacterial phyla likely also produce these lipids in nature (e.g., Sinninghe Damsté et al. 2011, 2018; Weber et al. 2018; De Jonge et al. 2019; van Bree et al. 2020; Sahonero-Canavesi et al. 2022; Halamka et al. 2023).



**Table 1.** Formulas of GDGT-based proxies used in this study. GDGTs in square brackets refer to the fractional abundances. The 6-Me brGDGTs are indicated with the prime symbol.

| Formula | Reference |
|---|---|
| $TEX_{86} = \frac{([GDGT\text{-}2]+[GDGT\text{-}3]+[cren'])}{([GDGT\text{-}1]+[GDGT\text{-}2]+[GDGT\text{-}3]+[cren'])}$ | Schouten et al. (2002) |
| $BIT = \frac{([Ia]+[IIa]+[IIa']+[IIIa]+[IIIa'])}{([Ia]+[IIa]+[IIa']+[IIIa]+[IIIa']+[crenarchaeol])}$ | Hopmans et al. (2004) |
| $f[CREN'] = \frac{[cren']}{([cren']+[crenarchaeol])}$ | Baxter et al. (2021) |
| $\%GDGT\text{-}2 = \frac{100*isoGDGT\text{-}2}{isoGDGT\text{-}2+isoGDGT\text{-}2+isoGDGT\text{-}3+cren']}$ | Sinninghe Damsté et al. (2012a) |
| $IR_{6Me} = \frac{IIa'+IIb'+IIc'+IIIa'+IIIb'+IIIc'}{IIa'+IIb'+IIc'+IIIa'+IIIb'+IIIc'+IIa+IIb+IIc+IIIa+IIIb+IIIc}$ | De Jonge et al. (2015) |
| $MBT'_{5Me} = \frac{([Ia]+[Ib]+[Ic])}{([Ia]+[Ib]+[Ic]+[IIa]+[IIb]+[IIc]+[IIIa])}$ | De Jonge et al. (2014) |
| $CBT' = log\frac{([Ic]+[IIa'])}{([Ia]+[Ib]+[Ic]+[IIa]+[IIb]+[IIc]+[IIIa])}$ | De Jonge et al. (2014) |
| $DC = \frac{([Ib]+2*[Ic]+[IIb]+[IIb'])}{([Ia]+[Ib]+[Ic]+[IIa]+[IIa']+[IIb]+[IIb'])}$ | Sinninghe Damsté (2016); Baxter et al. (2019) |
| $MST = 20.9 + 98.1*[Ib] - 12*([IIa]+[IIa']) - 20.5*([IIIa]+[IIIa'])$ | Pearson et al. (2011) |

BrGDGTs are particularly abundant in soils (Weijers et al., 2006b). The Branched versus Isoprenoid Tetraether (BIT) index
(Hopmans et al. 2004; Table 1), a ratio expressing the relative abundance of presumed soil-derived brGDGTs and aquatically
produced crenarchaeol, was initially used to assess the contribution of terrestrial material to the sedimentary GDGT pool in
coastal marine settings. Consequently, in lacustrine settings, the BIT index was also thought to track terrestrial organic matter
input into the lake system via soil erosion and runoff (Sinninghe Damsté et al., 2009; Blaga et al., 2009). However, it is now
established that *in situ* production of brGDGTs in lakes is significant and probably dominant in most systems (Tierney and
Russell, 2009; Sinninghe Damsté et al., 2009; Woltering et al., 2012; Weber et al., 2015, 2018; van Bree et al., 2020). More
recently, it has been shown that in certain stratifying lakes the BIT index may reflect long-term changes in lake depth, and
therefore is rather a proxy for integrated climatic moisture balance than of rainfall amount (Baxter et al., 2021).

Besides the BIT index, several other climate proxies based on brGDGT distributions have been developed. In essentially all
studied settings the distribution of these lipids displays a strong correlation to temperature, which following the discovery of
6-Me brGDGTs (De Jonge et al., 2014) is best reflected in the degree of methylation of 5-Me brGDGTs, as captured by the





methylation of branched tetraether ($\mathrm{MBT'_{5Me}}$) index (Table 1; De Jonge et al. 2014). Warmer climates produce a generally higher relative abundance of less methylated brGDGTs (Weijers et al., 2007; Raberg et al., 2022) and, hence, $\mathrm{MBT'_{5Me}}$ may be used to reconstruct past continental temperatures. The potentially mixed soil and aquatic origin of brGDGTs in lakes (e.g., Niemann et al. 2012; Naeher et al. 2014; Miller et al. 2018) originally created uncertainty about which calibrations are most ap-

propriate there, followed by clear support for the development of calibrations specifically applicable to lake sediments (Tierney et al., 2010b; Pearson et al., 2011; Russell et al., 2018; Martínez-Sosa et al., 2021; Raberg et al., 2021). Modern system studies continued to highlight the relative importance of lacustrine brGDGTs production (Sinninghe Damsté et al., 2009; Tierney and Russell, 2009; Bechtel et al., 2010) and application of a soil calibration to lake sediments produced temperature estimates differing ∼10 °C from observations (Tierney et al., 2010b). Despite the strong correlation between $\mathrm{MBT'_{5Me}}$ in lacustrine surface

sediments and temperature (Russell et al., 2018; Martínez-Sosa et al., 2021; Raberg et al., 2021), only few downcore applications of lake-based temperature calibrations since discovery of the 5-Me and 6-Me isomers have proved successful (Feakins et al., 2019; Stockhecke et al., 2021; Zhao et al., 2021; Zhang et al., 2021; Garelick et al., 2021; Ramos-Roman et al., 2022; Parish et al., 2023), partly due to uncertainties about the exact source(s) of brGDGTs in lakes. Recommendations to select temperature calibrations based on geographic region and/or mixing regime of the reconstruction site (Loomis et al., 2014b) are

not strictly followed, and some studies use modified indices that seem better suited to the particular study site or reconstruction (e.g., Bittner et al. 2022; Baxter et al. 2023). Clearly, application of brGDGT-based paleothermometers to lake-sediment archives shows great potential but is far from straightforward at present.

The influence of pH on the distribution of brGDGTs varies between environmental settings. Studies of soil and peat carried out both before (Weijers et al., 2007; Peterse et al., 2010, 2012) and after (De Jonge et al., 2014; Xiao et al., 2015; Naafs et al.,

2017a, b) the discovery of the 5-Me and 6-Me brGDGTs have shown that the degree of cyclisation of brGDGTs (represented by the cyclisation of branched tetraether index; CBT) and also the relative proportion of the 6-Me isomer (represented by the isomer ratio; $\mathrm{IR_{6Me}}$) are strongly related to pH. By contrast, the influence of pH on brGDGT distributions in lacustrine surface sediments is generally weaker than in other continental settings and sometimes reportedly absent (Loomis et al., 2014a; Russell et al., 2018; Martínez-Sosa et al., 2021; Raberg et al., 2022), possibly relating to the often large pH gradient with depth

and unresolved origin of the brGDGTs in most lakes. Moreover, in lake water microcosm experiments, no influence of pH on brGDGT distributions was found (Martínez-Sosa et al., 2020). Also, although culture studies found relationships between the methylation of the 5-Me brGDGTs and temperature mirroring those found in natural settings, the same studies showed no response of brGDGT distributions to pH, indicating that community change may be the primary driver of the sensitivity to pH observed in nature (Chen et al., 2022; Halamka et al., 2023). This is also supported by simulations of membrane dynamics,

which similarly did not find clear evidence that the cyclisation of brGDGTs is a means to control membrane fluidity in response to pH variation (Naafs et al., 2021). Moreover, due to the manner in which the commonly applied CBT' index is calculated (De Jonge et al., 2014), it does not only represent the degree of cyclisation, but is heavily influenced by differences in the relative abundance of the 5-Me and 6-Me isomers. Hence, some studies apply another method for calculating the degree of cyclisation of the brGDGTs (DC; Sinninghe Damsté 2016; Baxter et al. 2021).



Besides temperature and pH, several other environmental variables such as lake depth (Tierney et al., 2010b; Loomis et al., 2014a), nutrient availability (Loomis et al., 2014b; Martínez-Sosa and Tierney, 2019), conductivity (Shanahan et al., 2013; Raberg et al., 2021), and dissolved oxygen content (Loomis et al., 2014a, b; Martínez-Sosa and Tierney, 2019; van Bree et al., 2020; Yao et al., 2020; Wu et al., 2021) have been found to potentially impact brGDGT distributions in experimental and natural settings. Redox conditions, in particular, have shown to exert a significant influence on the concentration and distributions of

brGDGTs in lacustrine surface sediments (Loomis et al., 2014a; Wu et al., 2021), as well as their spatial pattern within the water column of some stratifying lakes (Weber et al., 2018; van Bree et al., 2020; Yao et al., 2020). This sensitivity to redox conditions has been further substantiated by micro- and mesocosm experiments (Martínez-Sosa and Tierney, 2019).

    In summary, most GDGT-based climate reconstructions from lake sediment records are supported only by region-specific or global calibrations relating the distribution of GDGTs in recently deposited lacustrine sediments to targeted climate variables.

To validate these empirical proxy-climate relationships, investigation of the modern lake system and depositional environment is crucial to identify the influence of confounding factors on the exact relationship of specific GDGTs with temperature or moisture balance. However, monitoring proxy variation in the modern system across seasonal to interannual time scales does not necessarily suffice to explain proxy variation at the much longer time scale of climate reconstruction. At this longer time scale, long-lived lakes are dynamic systems experiencing large-scale physical, chemical and biological changes throughout

their history related to the geological evolution of the lake basin since its formation, its gradual infilling with sediments, changes in the basin's hydrographic network and/or local tectonics. The influence of these long-term changes in the lake system on the local aquatic microbial communities may significantly impact the reliability of GDGT-based climate proxies, but regrettably this important source of uncertainty in the reliability of climate reconstructions is rarely discussed.

    For almost two decades, Lake Chala near Mt. Kilimanjaro has been the focus of lake monitoring and climate proxy validation

studies aimed at producing a long and robust paleoclimate reconstruction from eastern equatorial Africa with high temporal resolution. Particularly, a series of multi-year studies investigating the occurrence and climatic significance of various GDGTs (Sinninghe Damsté et al., 2009; Buckles et al., 2013, 2014; van Bree et al., 2020; Baxter et al., 2021) resulted in Lake Chala being ranked amongst the best studied lakes worldwide with regards to GDGTs. These modern-system investigations were carried out in preparation of the International Continental Scientific Drilling Program (ICDP) project DeepCHALLA (Verschuren

et al., 2013), which in 2016 extracted a 214.8-m sediment sequence of continuously laminated, diatom- and organic matter-rich sediments from Lake Chala covering the last c. 250 ka (Verschuren et al., 2013; Martin-Jones et al., 2020). At present, very few continuous climate records from tropical East Africa extend beyond the Last Glacial Maximum (LGM; c. 23,000 to 19,000 years ago), and even fewer cover the last glacial (MIS4–MIS2) or last interglacial (MIS5) periods (i.e., Tierney et al. 2008; Loomis et al. 2012; Johnson et al. 2016. While the long sediment sequence recovered by DeepCHALLA represents a unique

opportunity to extend detailed knowledge of East African climate history back to the previous glacial-interglacial cycle (MIS6-MIS7), achieving this objective is contingent on confirmation that our understanding of the relationship between selected sedimentary proxies and climate is applicable throughout the record. To reach this objective, the present study aims to relate the distributions of isoGDGTs, brGDGTs and the associated climate-proxy indices in 949 horizons throughout the c. 250-kyr DeepCHALLA sediment sequence to major phases in the basin evolution and aquatic ecology of Lake Chala as revealed by





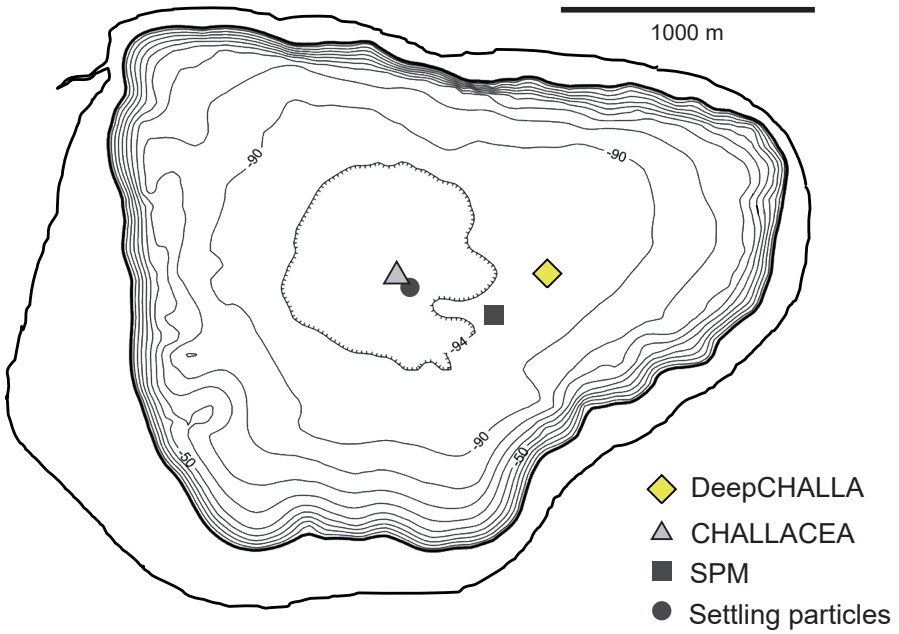

**Figure 1.** Bathymetry of Lake Chala (adapted from Moernaut et al. (2010) with the outer bold line demarcating the crater catchment. The drilling sites of the DeepCHALLA and CHALLACEA campaigns are indicated by a yellow diamond and grey triangle, respectively. Also shown are the fixed sampling locations of suspended particulate matter (SPM; dark grey square) and settling particles (dark grey circle) used in earlier studies (Sinninghe Damsté et al., 2009; Buckles et al., 2014; van Bree et al., 2020; Baxter et al., 2021).

independent paleoenvironmental proxies derived from seismic reflection data, lithology and fossil diatom assemblages). We build on our understanding of GDGT proxy-climate relationships informed by the modern-system and more shallow sediment studies to examine the stability of these relationships throughout deposition of the entire DeepCHALLA record. This integrated analysis enabled us to identify and disentangle the influences of lake development and climate variability on the concentrations and distributions of GDGTs in the DeepCHALLA sedimentary archive.

## 2   Study site and results of previous work

### 2.1   Site description

Lake Chala (also spelled 'Challa' after the nearby village) is a relatively small (4.2 $km^2$), deep (c. 90 m in 2016) volcanic crater lake bridging the border of Kenya and Tanzania in eastern equatorial Africa (3°19'S, 37°42'E), situated at c. 880 m above sea level in the southeastern foothills of Mt. Kilimanjaro (Fig. 1). Lake-surface evaporation (1735 mm $yr^{-1}$; Payne (1970)) greatly



exceeds average annual rainfall (565 mm yr$^{-1}$; De Wispelaere et al. 2017; Griepentrog et al. 2019). Therefore, besides rainfall on the lake and on the steep inner slopes of the crater basin, substantial subsurface inflow is required to balance the lake's water budget (Payne, 1970). This subsurface inflow is derived from percolating rainfall on or above the forested slopes of Mt. Kilimanjaro (Hemp, 2006; Bodé et al., 2020) that reaches the lake 3–4 months later (Barker et al., 2011a). Presently, Lake Chala is a fresh, slightly alkaline (surface-water pH 8.4-9.3) and unproductive lake with high concentrations of silica but low

concentrations of phosphorus and nitrogen in the mixed surface layer, although these nutrients accumulate in the hypolimnion (Wolff et al., 2014). The lake is topographically closed but occasionally after high rainfall a small creek is activated, which breaches the north-western crater rim (Buckles et al., 2014). The lake has a typical crater-lake morphology, with steep crater walls that reach up to 170 m above the lake's surface and steep underwater slopes down to ∼60–70 m which level off to form a flat central lake bottom (Moernaut et al., 2010). It has a roughly triangular shape with a total catchment area (5.6 km$^2$) that

is only 30% larger than the surface area of the lake itself. From approximately 10 m above the 2016 lake level, i.e. the upper limit of shallow caves formed by wave erosion during past high-stands, more significant outflow is possible through the porous upper crater walls.

In this semi-arid tropical climate regime, highest mean monthly air temperatures are reached in February–March (night and daytime temperature of 21 and 33 ℃), and lowest temperatures in July–August (night and daytime temperature of 18 and 28

℃; Buckles et al. 2014). Lake Chala is located east of the Congo Air Boundary (CAB) and is therefore orographically isolated from Atlantic- or Congo Basin-sourced moisture (Sepulchre et al., 2006; Verschuren et al., 2009; Tierney et al., 2013). It is located in the so-called greater Horn of Africa region which is drier than more western parts of the continent at comparable latitude due to relatively modest rainfall from the Indian Ocean. The region's climate is characterized by a bimodal pattern of seasonal rainfall (Wainwright et al. 2019 and references cited therein) associated with the shifting latitudinal position of the

intertropical convergence zone (ITCZ) and tropical rain belt. Short and long rains occur from late October to December and from March to May, respectively. They are separated by the main dry season from June to September, i.e. during the southern hemisphere (SH) winter, and a short dry season in January-February.

## 2.2  Water-column depth zones and mixing regime

Following Buckles et al. (2014), the water column of Lake Chala can be separated into 6 distinct zones (Fig. 2), differentiated

by their frequency of mixing as reflected in physical and chemical properties. Zone 1 represents the daily mixed layer, and is fully oxygenated with uniform temperature and pH. Zone 2 is characterized by oxic to sub-oxic conditions and positioned from immediately below the principal thermocline to the oxycline which demarcates its base. Zone 3 is thus anoxic with pH falling to c. 7.2 and sharply increasing concentration of dissolved methane which continues to increase through the lower water column (Baxter et al., 2021). Zones 1–3 together constitute the mixolimnion, i.e., the portion of the water column that mixes

at least once each year (Lewis, 1983; De Crop and Verschuren, 2021). Zones 4–6 together constitute the monimolimnion, and are defined by a stable temperature of 22.3 °C and permanent anoxia, except that rare deep-mixing events reaching into Zone 4 may occasionally inject oxygen that is however quickly consumed by bacterial activity. Across Zone 4, pH decreases further to c. 7.0 and the dissolved-ion concentration (measured as specific conductance) increases with depth from c. 350 µS/cm to c.





450 µS/cm (Barker et al., 2013; Wolff et al., 2014), creating a chemical density gradient across Zone 4 which largely prevents
temperature-driven convective mixing (and oxygen injection) beyond the Zone 3-4 boundary (De Crop and Verschuren, 2021).
Stable pH and dissolved-ion concentrations throughout Zone 5 indicate lack of mixing even on multi-annual time scales. This
also applies to Zone 6, but being positioned directly above the profundal lake bottom the local water chemistry and redox
conditions are affected by diffusion out of the uncompacted surficial sediments subject to diagenesis.

Lake Chala is characterized by a strong seasonal mixing pattern relating to the oscillation between windy dry and calm
wet seasons, with substantial variability in the expression of these seasons between successive years. From September until
May, i.e., the period encompassing the long and short rain seasons and the intervening warm dry season, high lake-surface
temperatures and/or lower wind speeds result in reduced mixing of the upper water column, promoting stronger temperature
and chemical stratification (Fig. 2a). Except for the daily mixed layer (Zone 1) oxygen renewal is diminished, and due to
heterotrophic bacterial activity that is being promoted by high water temperatures, the depth range of sub-oxic transitional
conditions (Zone 2) is greatly reduced or even eliminated, thereby shifting the oxycline (top of Zone 3) to a shallower position
(∼10 to 15 m), with most intensely stratified conditions occurring during SH summer (Wolff et al., 2014; van Bree et al., 2018;
van Bree et al., 2020). This expansion of the anoxic (but seasonally mixing) Zone 3 increases the overall volume of anoxic
water (Zones 3–6) relative to the oxygen-rich surface layers (Zones 1–2).

During the main dry season, lower air temperatures and higher wind speeds cause turbulent and convective deep mixing of
the upper water column to 42–46 m (Wolff et al. 2014; Buckles et al. 2014; van Bree et al. 2020; Fig. 2b). During this deep-
mixing period, which normally starts at the end of May and finishes by mid-September (Wolff et al., 2014; van Bree et al.,
2018), oxygen penetrates further into the water column causing the expansion of the oxygenated zones, most dramatically
of Zone 2. Consequently, the depth range of Zone 3 is greatly reduced, and nutrient-rich deep water is brought up to the
nutrient-starved epilimnion, promoting phytoplankton productivity (for example causing a pronounced diatom bloom; Wolff
et al. 2014; van Bree et al. 2018). Hence, the oxygenated portion of the water column (Zones 1–2) increases relative to the
anoxic portion (Zones 3–6). Also, during the short dry season in January–February, a period of shallower mixing to c. 20–25
m interrupts the long period of stratification, driven by high wind speeds but hampered by high surface-water temperature
(Wolff et al., 2014; van Bree et al., 2018; van Bree et al., 2020). To generalize, there are two seasonal extremes of mixing states
in Lake Chala associated with shallow or deep oxycline conditions, experienced during the mostly calm/wet and windy/dry
seasons, respectively. The permanent anoxia of the lower water column of Lake Chala leads to the deposition of organic-rich,
diatomaceous and often seasonally laminated sediments (i.e., varves; Wolff et al. 2011).

## 2.3  Depositional history of Lake Chala inferred from seismic stratigraphy

Seismic profiling of the crater basin of Lake Chala revealed that the lake overlies at least c. 210 m of near-continuous lacustrine
sedimentation (Moernaut et al., 2010). Based on the extrapolation of long-term mean sedimentation rates from the upper
portion of the seismic sequence (Moernaut et al., 2010) to the lowermost section of the seismic stratigraphy, it was inferred that
the sediments may reflect 250-kyr of deposition, i.e., the entirety of the two most recent glacial-interglacial cycles (Verschuren
et al., 2013). Detailed analysis of high-resolution seismic profiles of the Chala crater basin has permitted reconstruction of the





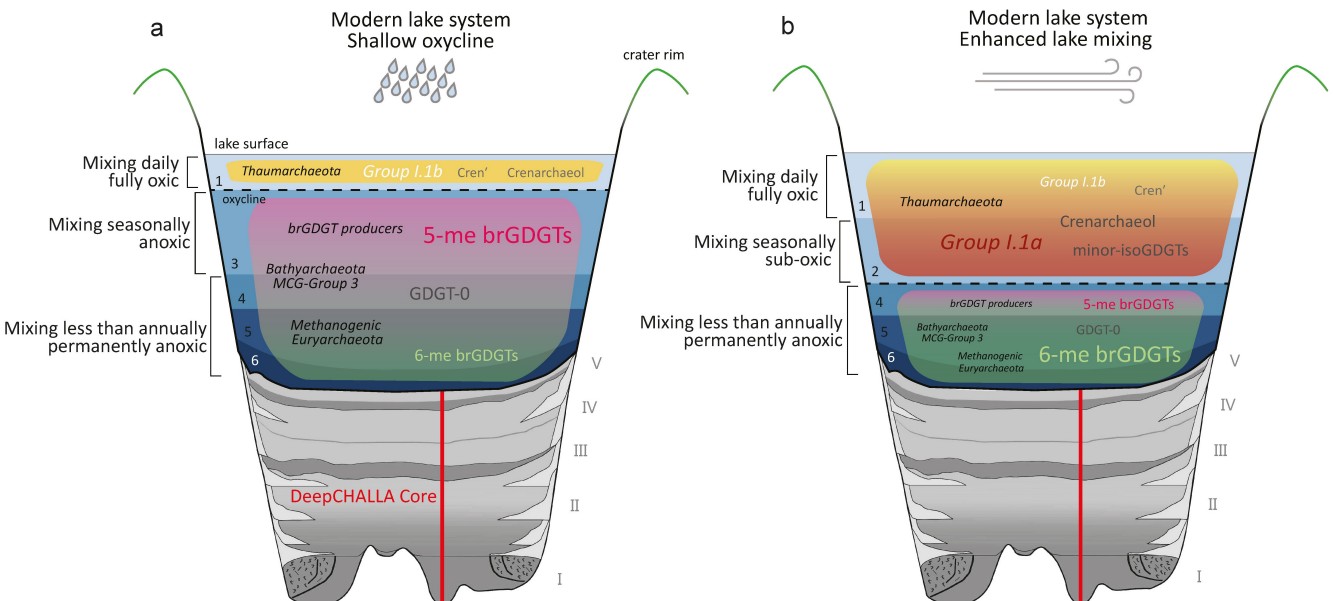

**Figure 2.** Schematic representation of the spatial distributions of GDGT producers in the modern Lake Chala water column during the two seasonal extremes of (a) highly stratified (shallow oxycline) conditions associated with the rainy season and (b) enhanced upper water-column mixing associated with windy/dry conditions based on distributions in SPM and settling particles (Buckles et al., 2013, 2014; van Bree et al., 2020; Baxter et al., 2021) and seismic profiling of the underlying sediments with the five major depositional stages (I-V) indicated (Maitituerdi et al., 2022). The lake is divided into six distinct mixing zones with different mixing frequencies and chemical properties, following Buckles et al. (2014). See sections 2.1 and 2.2 for further description.

complete depositional history of the lake back to 250 ka, beginning with the initial lacustrine sediment infill after the collapse of the caldera (Maitituerdi et al., 2022). Building on the earlier analysis of the upper half of the seismic profiles (Moernaut

et al., 2010), the stratigraphic sequence is characterized by either undisturbed and uniform basin-wide ('draped') sedimentation deposited under mostly high lake-level conditions, or basin-focused ('ponded') sedimentation reflecting periods of low lake level. The seismic record is hence separated into five depositional stages (Stages V–I), five successive phases in the basin's evolution characterized by pronounced changes in lake depth. Importantly, because ponded sedimentation reflects the greater sediment focusing which results when turbulent mixing extends to the profundal lake bottom (Maitituerdi et al., 2022), and

the steep-sided morphology of Chala basin entails constancy through time in the minimum water depth allowing undisturbed accumulation soft organic sediments (Håkanson and Jansson, 1983), seismic stratigraphy serves as a lake-level proxy tied more strictly to absolute water-column depth than to water-column structure and mixing regime. On this long time scale, lake level is a reflection of both climate-driven moisture variability and changes in basin hydrology as the crater progressively fills with sediments. Stage I (c. 248–207 ka), the oldest depositional phase, represents the initial lacustrine sedimentation which was

limited to a ring-shaped depositional area surrounding the then still exposed cones of volcanic tuff in the center of the basin,



and further characterized by thick mass-wasting deposits at the basin periphery (Maitituerdi et al., 2022). Lake level during Stage I is estimated to be low, but this could either be due to relatively dry regional climate conditions or the still leaky nature of the crater basin during the period following caldera collapse. Stage II (c. 207–113 ka) is defined by the complete burial of the central tuff cones and the start of basin-wide sedimentation. During the first half of Stage II (c. 207–147 ka) a gradual

transition to more draped sedimentation takes place, indicating that the water column became progressively taller; due to the still relatively thin underlying sediment package, the greatest overall depth of the lake during its entire 250-kyr history was likely reached during the second half of this stage. Stage III (c. 113–99 ka) represents a distinct period of significantly reduced lake level implying severe climatic drought, as indicated by strongly basin-focused sedimentation in the seismic profiles. During Stage III, Lake Chala developed the flat central lake floor which still exists today, and consequently from this stage onwards

the total depositional area of the crater basin has remained fairly constant through time. Stage IV (c. 99–19 ka) was a period of mostly high lake-level conditions, as implied by continuity of draped sedimentation, except for a short-lived low-stand c. 60 ka ago. Lastly, Stage V (19 ka to present) represents a period of fluctuating lake level, as inferred from a succession of ponded lenses reflecting sediment focusing under reduced lake level (Maitituerdi et al., 2022).

## 2.4 Ecological history of Lake Chala inferred from fossil diatom assemblages

Lake Chala sediments contain abundant fossil diatoms (Wolff et al., 2011, 2014), and consequently the analysis of diatom assemblages in the DeepCHALLA sequence has provided insights into upper water-column mixing, water chemistry, and nutrient dynamics throughout the lake's 250-kyr history (Tanttu, 2021). The pelagic (open-water) diatom community of modern-day Lake Chala is relatively poor in species, consisting mainly of *Afrocymbella barkeri* (Cocquyt and Ryken, 2016) and *Nitzschia fabiennejansseniana* (Cocquyt and Ryken, 2017). The more common presence of diatom species associated with littoral (near-

shore) and benthic (bottom) habitats during the early history of the lake (c. 248–221 ka) suggests that Lake Chala at that time was a relatively shallow and well-mixed lake environment with greater nutrient availability. Fossil diatom assemblages deposited after c. 221 ka are entirely dominated by *N. fabiennejansseniana* and other needle-shaped *Nitzschia* species, indicating a change to a purely pelagic environment with deep water column, weak upper water-column turbulence, and strongly nutrient-limited conditions. *A. barkeri* makes its first appearance c. 144 ka, and from then on the diatom community had a composition

similar to that of the modern-day lake, except for major temporal variability in dominance of one or more needle-like *Nitzschia* spp. aversus the more heavily silicified *A. barkeri*. Dominance of the latter is interpreted to reflect episodes with greater mixed-layer turbulence and more efficient nutrient recycling, whereas the former reflects a shallower or weaker mixed upper layer and therefore more extreme nutrient depletion. To the extent that alternation between these conditions was climate-controlled, *A. barkeri* dominance may therefore be associated with reduced lake depth (lower lake level) during periods of drier climate

conditions (Tanttu, 2021). *A. barkeri* dominated the diatom community of Lake Chala during the periods dated to 108–96 ka and 28–13 ka. From 94 ka to 48 ka, reappearance of *Nitzschia* suggests a transition towards greater lake depth and more stable upper water column. Importantly, the fossil diatom data indicate that Lake Chala has been a true freshwater lake throughout its 250-kyr history (Tanttu, 2021).





## 2.5 GDGTs in the modern Lake Chala system, and climate-proxy relationships

The aim to better understand the producers, sources, and climatic sensitivity of GDGTs extracted from the DeepCHALLA sediment record, and thus to validate the climate-proxy relationships used for paleoclimate reconstruction, has stimulated extensive investigations into the occurrence and distribution of specific GDGTs in Lake Chala and its surrounding catchment (Sinninghe Damsté et al., 2009; Buckles et al., 2013, 2014, 2016; van Bree et al., 2020; Baxter et al., 2021). To determine seasonal to interannual-scale variation in GDGT distributions, some of these studies involved monthly monitoring of lake

conditions and measurement of GDGT variation spanning multiple years. Namely, brGDGTs (van Bree et al., 2020) and isoGDGTs (Baxter et al., 2021) were analyzed in settling particles collected at 35 m water depth during 98 consecutive months, and in suspended particulate matter (SPM) collected at 13 discrete water depths during 17 consecutive months (Fig. 1). This section provides an overview of the outcome of these studies for the interpretation of GDGT proxies, to later be applied in the context of the 250-kyr DeepCHALLA sedimentary record.

### 2.5.1 The BIT index


In the 25-kyr record from the CHALLACEA site in Lake Chala (Fig. 1), the BIT index showed good agreement with a first-order lake-level reconstruction based on seismic stratigraphy (Verschuren et al., 2009). It was also recognized that soils in the hills surrounding Mt.Kilimanjaro contain high amounts of brGDGTs (Sinninghe Damsté et al., 2008). Hence, based on the then-current understanding of brGDGT sources in lakes, the BIT index was inferred to reflect varying transport of soil-derived

brGDGTs to the lake associated with varying precipitation and consequent soil erosion. Following the discovery that brGDGTs in Lake Chala are abundantly produced within the lake itself (Buckles et al., 2013, 2014), additional modern-system studies elucidated the exact nature of the relationship between the Chala BIT index and hydroclimate (van Bree et al., 2020; Baxter et al., 2021). Namely, brGDGTs are primarily produced in the anoxic zone of the water column (Zones 4–6), and their depth range follows the seasonal cycle of lake mixing and stratification such that during the deep-mixing season between June and

September they are restricted deeper in the water column, while under conditions of strong upper water-column stratification during the rest of the year, the oxycline moves upwards, thereby expanding the brGDGT production zone (Fig. 2; van Bree et al. 2020). On the other hand, the niche of Group I.1a Thaumarchaeota, generally the main producers of crenarchaeol in Lake Chala (with secondary contributions from Group I.1b; Buckles et al. 2013; Baxter et al. 2021), is primarily restricted to the (sub-)oxic zone between the principal thermocline and the oxycline (Zone 2), where the degree of sunlight is much less intense

than in the uppermost layer and ongoing or recent deep mixing provides ammonium (Buckles et al. 2013; Baxter et al. 2021; Fig. 2a). During periods of prolonged shallow oxycline conditions, this depth niche of Group I.1a Thaumarchaeota (Zone 2) is eliminated, and their annual "bloom" is suppressed (Buckles et al. 2013, 2014; Baxter et al. 2021; Fig. 2b). In a study of SPM sampled throughout the water column during a year when exceptionally shallow oxycline conditions prevailed, only gene copies of Group I.1b Thaumarchaeota were detected and crenarchaeol concentrations were several orders of magnitude lower

than recorded previously (Buckles et al., 2016; Baxter et al., 2021). With substantially lower amounts of crenarchaeol settling on the lake bottom during such intervals, the accumulating sediments attain higher BIT-index values (Baxter et al., 2021).



During periods of sustained deep mixing, the reverse situation of prolonged oxygenation of the upper water column (Zones 1–2) promotes development of Thaumarchaeota, thus increasing crenarchaeol production and lowering BIT-index values.

Therefore, the BIT index effectively tracks changes in the relative size of the anoxic and oxygenated zones in the water
column in Lake Chala. Within a single year, the oxycline position is controlled by the timing and duration of seasonal deep mixing related to monsoon variability, such that when the intertropical convergence zone (ITCZ) is overhead, heavy rainfall and low wind speeds cause shallow oxycline conditions, and when the ITCZ is located to the North/South, high wind speeds and lack of rainfall enhance lake mixing (van Bree et al., 2020; Baxter et al., 2021). On the long time scales of paleoclimate reconstruction, the relative proportion of the anoxic and oxic zones will also be strongly influenced by changes in overall lake
depth (De Crop and Verschuren, 2021). High-stand episodes of greater lake depth will be associated with an overall taller anoxic zone, whereas during low-stands the anoxic zone will shrink, increasing the relative volume of the upper mixed layer (Verschuren, 1999, 2001). Hence, on long time scales registered in Lake Chala sediments, the BIT index is a reflection of temporal variation in hydrological moisture balance (Baxter et al., 2021), which in this relatively dry tropical region is chiefly determined by changes in the strength of the Indian Monsoon and temperature effects on continental evaporation (Baxter et al.,
2021, 2023).

### 2.5.2  IsoGDGT distribution and TEX$_{86}$

The ephemeral nature of Zone 2 in the water column of Lake Chala, where Group I.1a Thaumarchaeota are most abundant, has a major influence on sedimentary proxies based on isoGDGTs (Baxter et al., 2021). During periods of exceptionally shallow oxycline and Thaumarchaeotal bloom suppression, greater contributions from methanotrophs, methanogens and other
anaerobic archaea to the isoGDGT pool render the temperature signal derived from TEX$_{86}$ untrustworthy (Sinninghe Damsté et al., 2012a; Baxter et al., 2021). In this way, temporal variation in mean annual water-column stratification is a crucial factor controlling the TEX$_{86}$ signal in Lake Chala, potentially equally important as temperature variation (Baxter et al., 2021), in line with the results of other modern system studies (e.g., Zhang et al. 2016; Cao et al. 2020; Dang et al. 2016; Sinninghe Damsté et al. 2022). As methanogens produce relatively high amounts of isoGDGT-0, the ratio between isoGDGT-0 and crenarchaeol
(isoGDGT-0/cren) has been used to assess the contribution of methanogens to the sedimentary isoGDGT pool (e.g., Blaga et al. 2009; Bechtel et al. 2010). Similar to brGDGTs, also isoGDGT-0 is produced most abundantly in the anoxic lower water column of Lake Chala (Buckles et al., 2013; Baxter et al., 2021). Therefore, the isoGDGT-0/cren ratio likewise reflects changes in the relative volume of the anoxic and oxic portions of the water column, and is relatively higher during highly stratified lake conditions and lower during periods of deep mixing (Fig. 2). A greater relative abundance of the crenarchaeol isomer
may reflect periods during which the contribution of Group I.1b Thaumarchaeota to the isoGDGT pool is increased, as these archaea produce a greater amount (typically 14–29%) of the isomer than Group I.1a Thaumarchaeota (typically only 0–3%; Pitcher et al. 2010, 2011; Kim et al. 2012; Sinninghe Damsté et al. 2012b; Elling et al. 2017; Bale et al. 2019). Significantly, Group I.1b Thaumarchaeota do not produce isoGDGTs with the same temperature dependency of ring formation as Group I.1a (e.g., Elling et al. 2017). In Lake Chala, similarly to crenarchaeol itself, the crenarchaeol isomer is most abundant in the
oxygenated Zones 1–2 (Fig. 2). In the 98-month data set of settling particles, higher than average f[CREN′] values (a measure



of the contribution of the crenarchaeol isomer; Table 5.1) were recorded during a sustained period in 2013 when only Group I.1b Thaumarchaeota gene copies were detected (Baxter et al., 2021). Hence, it appears that the two groups of Thaumarchaeota are differentially impacted by lake stratification, with Group I.1a being severely diminished when the (sub-) oxic Zone 2 is eliminated or reduced (Fig. 2b). The f[CREN′] proxy can therefore be used as an indicator of prolonged shallow-oxycline conditions in Lake Chala (Baxter et al., 2021).

### 2.5.3 BrGDGT distribution and associated proxies

The 17-month time series of SPM data from Lake Chala revealed that the 5-Me and 6-Me brGDGT isomers appear to occupy spatially distinct zones within the anoxic lower water column (van Bree et al., 2020). Namely, the 5-Me brGDGTs are produced mainly in the anoxic but seasonally variable Zone 3, whereas 6-Me brGDGTs are produced most abundantly in the equally anoxic but permanently stratified Zones 4–6 (Fig. 2). As presented above, Zone 3 is greatly reduced during periods of deep lake mixing, hence limiting the growth of 5-Me brGDGT producers and increasing the relative contribution of 6-Me brGDGT producers. The isomer ratio ($IR_{6Me}$; Table 5.1) captures this relative contribution of 6-Me to 5-Me GDGTs. Indeed, low $IR_{6Me}$ values in settling particle data (van Bree et al., 2020) correspond to trends in other proxies (BIT index, isoGDGT-0/cren) indicative of unusually shallow oxycline conditions (Baxter et al., 2021). Just as in the case of the BIT index, on the long timescales reflected in sedimentary records, changes in the $IR_{6Me}$ ratio will be predominantly controlled by changes in lake depth, because the increased inputs of fresh water which cause lake level to rise lead to the expansion of Zone 3 where 5-Me brGDGT producers proliferate, hence causing lower $IR_{6Me}$ values in the sediment (Baxter et al., 2023). The latter study concluded, supported by findings from detailed water column studies in Lake Chala (van Bree et al., 2020; Baxter et al., 2021), that temperature calibrations based on $MBT'_{5Me}$ are not suitable for local paleotemperature reconstruction because past episodes of low lake level likely resulted in strong reduction of Zone 3 where the 5-Me brGDGT isomers are produced. Instead it was shown that 6-Me brGDGTs need to be included in the transfer function to properly capture the temperature sensitivity of the local brGDGT producers.

## 3 Materials and Methods

### 3.1 Construction of the sediment sequence, lithofacies description and age model

In 2016 the International Continental Scientific Drilling Program (ICDP) project 'DeepCHALLA' recovered a sedimentary sequence of c. 214.8 m below the lake floor by hydraulic piston coring (Fig. 1). Drilling occurred in five main drill holes (A–E) at a single location in the eastern depocenter of Lake Chala, with overlapping 3-m sections achieving complete (100%) recovery in the upper 123 m (c. 160 ka to present) of the sediment sequence and near-complete (∼85%) recovery of the lower 92 m.

The splitting, imaging, non-destructive scanning, preliminary lithological description, and cross-correlation of the DeepCHALLA cores were carried out at the U. S. National Lacustrine Core Facility (LacCore) hosted by the University of Minnesota (Min-



neapolis, USA), as previously described by Baxter et al. (2023) with respect to the upper 68.39 m of the composite sediment sequence. The entire drilled sequence of these matrix sediments consists of fine-grained and diatom-rich organic muds with visually clear mm-scale lamination or cm-scale banding. This allowed overlapping core sections from different drill holes to be precisely cross-correlated at the mm-scale while viewing the high-resolution digital line-scan images in Corelyzer software, and identifying either shared lamination features or the base and top of turbidites as robust stratigraphic tie points. Before extraction of sediment samples for analysis of bulk sediment composition, organic biomarkers and fossil diatom assemblages, among others, all event deposits (turbidites) with thickness >2 cm were excluded from the continuous depth scale, to obtain a provisional 'event-free' depth scale and to ensure that samples mostly reflect genuine 'matrix' sediments of primary lacustrine deposition at the drill site. Sets of samples were extracted from the work halves of core sections represented in the composite sequence, at predetermined constant depth intervals on the event-free depth scale such that proxy time series have a more or less uniform temporal resolution throughout the sediment record. Following detailed inventory of all turbidites (Swai, 2018) the event-free depth scale was updated to also exclude turbidites of 0.5–2.0 cm thickness, prompting exclusion of some already analysed samples from the final proxy time series (see below).

Absolute dating efforts of the DeepCHALLA sequence are ongoing. Considering the focus of the present study on long-term lake-basin development rather than paleoclimate reconstruction, we use a preliminary sediment chronology based on transfer of the high-resolution $^{14}$C-based age model for the last 25 ka at the CHALLACEA site (Blaauw and Christen, 2011) to the DeepCHALLA site; links between the seismic stratigraphy of Chala basin at both sites and known near-global climate events back to 140 ka (Moernaut et al., 2010; Maitituerdi et al., 2022); and linear extrapolation of the average sedimentation rate over this 140-ka interval to the base of the DeepCHALLA core sequence at 215 m below the lake floor (Martin-Jones et al., 2020).

During sampling of the DeepCHALLA sequence in June 2017 the general appearance of the sediment at each 2-cm thick sampled depth interval was noted, with reference to the preliminary lithological description executed at LacCore. Matrix sediments were classified into one of two primary lithofacies, namely mm-scale (varve-like) laminations and cm-scale (banded) sediments. A third lithofacies type is used to describe core sections characterized by rapid alternation between these two facies at the cm-scale (Baxter et al., 2023). Mm-scale laminated sediments are interpreted to have been deposited under stable stratification and a permanently anoxic lower water column as exists in the lake today, whereas cm-scale banding reflects post-depositional disturbance of the uppermost few cm of originally mm-scale laminated muds due to bottom currents associated with occasional complete water-column mixing. Although such events may have injected some oxygen to the near-bottom environment, almost certainly this must have been consumed rapidly (in days rather than weeks) by bacterial activity (Lewis, 1987; De Crop and Verschuren, 2019) such that for all practical purposes the lower water column would still have been permanently anoxic. Nevertheless, sections of mm-scale lamination can be considered to represent periods of stable meromixis, whereas cm-banded sections represent periods complete water-column mixing occurred at least occasionally at the scale of decades.



## 3.2 Sample preparation and analysis of GDGTs

The present study involved a total of 949 sediment horizons from throughout the DeepCHALLA sequence, each 2 cm thick and extracted at a regular interval of 16 cm in matrix sediments (i.e., skipping turbidites >2 cm thick). Detailed inventory of all turbidites (Swai, 2018) revealed that 73 sediment horizons extracted and analyzed for GDGTs (7.5% of the total) partly consist of thin turbidites (< 2 cm thick). Here only the 33 samples containing >25% of turbidite material (3.3% of the total) were excluded from the final proxy time series, which hence consist of 916 sediment horizons spanning the past c. 250,000

430 years. Methods of sample preparation and GDGT analysis on DeepCHALLA sediments have been described previously (Baxter et al., 2023). In short, freeze-dried and powdered sediments (0.3–1.2 g dry weight) were extracted with a Dionex accelerated solvent extraction (ASE) system using a 9:1 v/v mixture of dichloromethane (DCM) and methanol and 1 $\mu$g of internal standard (synthetic C46 glycerol trialkyl glycerol tetraether; GTGT) was added to the total lipid abstract (TLE) (Huguet et al., 2006). TLEs were dissolved in DCM, passed through a $Na_2SO_4$ column and dried under $N_2$ gas before being separated into apolar,

ketone and polar fractions using eluents of hexane/DCM (9:1, v/v), hexane/DCM (1:1, v/v), and DCM/methanol (1:1, v/v), respectively, and passing through an $Al_2O_3$ column. The fractions were dried under $N_2$ gas and the polar fractions, containing the GDGTs, were redissolved in hexane/isopropanol (99:1, v/v) prior to being filtered using a PTFE 0.45 $\mu$m filter. Measurement of GDGTs was carried out using an Agilent 1260 Infinity ultrahigh performance liquid chromatography (UHPLC) system coupled to an Agilent 6130 single quadrupole mass detector following the method of (Hopmans et al., 2016). GDGTs were

identified by $[M + H]^+$ ion detection in selected ion monitoring (SIM) mode for m/z 1018.0, 1020.0, 1022.0, 1032.0, 1034.0, 1036.0, 1046.0, 1048.0, 1050.0 (brGDGTs), m/z 1292.3, 1294.3, 1296.3, 1298.3, 1300.3 and 1302.3 (isoGDGTs), and m/z 743.8 (internal standard) with a mass window of 1.0. Peak area integration of the peaks representing GDGTs in the $[M + H]^+$ mass chromatograms was done using Agilent Masshunter software. A peak area of 3 $*10^3$ units was used as the detection threshold, with peaks below this threshold being excluded for proxy calculation.

## 3.3 Calculation of GDGT concentrations and derived climate proxies

Absolute concentrations of isoGDGTs and brGDGTs were normalized to the organic carbon ($C_{org}$) content of the sampled intervals and hence expressed in $\mu$g g$^{-1}$ $C_{org}$. Determination of $C_{org}$ was based on samples of c. 1.0 g of wet sediment representing the same 2-cm core increments as the samples extracted for GDGT samples. They were weighted immediately after extraction and again after freeze-drying to measure the loss in mass as estimate of water content (%$H_2O$). The freeze-dried

samples were homogenized, split in two and analyzed using a Primacs Carbon Analyzer at the University of Haifa (Israel), which determines total carbon content by combusting the sample at 1050 °C and measuring the evolved carbon dioxide. One subsample was treated with phosphoric acid before the measurement, to remove any inorganic carbon present and thus measure $C_{org}$ only (Maitituerdi, 2023). The GDGT concentration time series comprises 909 sediment horizons (as opposed n = 916 for the full biomarker proxy series) due to missing dry sample weight or %$C_{org}$ values for a handful of samples. The $TEX_{86}$ index

was calculated according to Schouten et al. (2002) (Table 1). The BIT index was calculated according to Hopmans et al. (2004) and modified to explicitly show the inclusion of both the 5- and 6-Me brGDGTs (De Jonge et al., 2014). In addition, isoGDGT-





0/crenarchaeol, f[CREN]′, and %isoGDGT-2 were calculated to investigate the producers contributing to the isoGDGT pool in Lake Chala (Sinninghe Damsté et al., 2012a; Baxter et al., 2021). $IR_{6Me}$ captures the relative abundance of 6-Me versus 5-Me isomers, with the 6-Me isomers indicated by the prime symbol, and was calculated according to (De Jonge et al., 2015). The methylation of 5-Me branched tetraether index ($MBT'_{5Me}$) and cyclisation of branched tetraether index (CBT') were calculated according to De Jonge et al. (2014). Additionally also the degree of cyclisation (DC) of brGDGT was calculated (Sinninghe Damsté, 2016; Baxter et al., 2021). For temperature reconstruction we applied the global lake calibration of Pearson et al. (2011), which calculates mean summer temperature (MST) and was determined by Baxter et al. (2023) to be the brGDGT-based paleotemperature-inference model best suited for application to Lake Chala sediments at the intended time scale. In Table 1, the original calibration is rewritten to highlight the inclusion of both 5-Me and 6-Me isomers, with GDGTs in square brackets referring to the fractional abundances. The resulting 250-kyr MST record was then rescaled to the mean temperature range of an ensemble temperature reconstruction for the last 25 kyr based on seven independent GDGT-based temperature reconstructions from other East African lakes, according to Baxter et al. (2023).

## 3.4 Numerical and statistical analysis

The relationships between temporal variation in individual GDGT compounds and selected proxies throughout the DeepCHALLA sediment sequence were explored using univariate and multivariate analyses in the R statistical package FactoMineR (Lê et al., 2008). Univariate analyses were performed on organic-matter normalized absolute concentrations, whereas multivariate principal component analyses (PCAs) were performed on the fractional abundances of individual GDGTs, relative to either the full suite of iso- and brGDGTs (22 compounds) or only the brGDGTs (15 compounds), isoGDGTs (7 compounds) or the sub-set of four isoGDGTs used in $TEX_{86}$ calculation (isoGDGT-1,-2,-3 and cren′). To assess the influence of lake-basin development and water-column mixing regime on GDGT distributions, and by extension their sensitivity to climate variability, the 916 analysed sediment horizons were grouped according to seismic lake-history stage or lithofacies to explore trends in GDGT concentrations, PCA scores, proxies, and possible relationships to changes in lake properties through time. Tukey multiple comparison of means with a 95% family-wise confidence interval was used to test if the GDGT concentrations and proxy values of these groups are significantly different; means are considered significantly different according at the 5% level of significance.

## 3.5 Periodicity analysis

The time series of GDGT concentrations and derived proxies were subjected to periodicity analysis using Acycle 2.3.1 software (Li et al., 2019), after standardization (mean = 0, standard deviation = 1) and 3rd-order detrending. For wavelet (morlet) analysis, the standardized and detrended GDGT time series were first resampled at the median temporal resolution (between 206 and 200 years depending on the selected time interval). Gaussian band pass filtering was also performed on select GDGT proxies using bandwidths that targeted periodicities compatible with obliquity (41-kyr) and precession (23-kyr) orbital insolation forcing, as revealed by wavelet analysis.



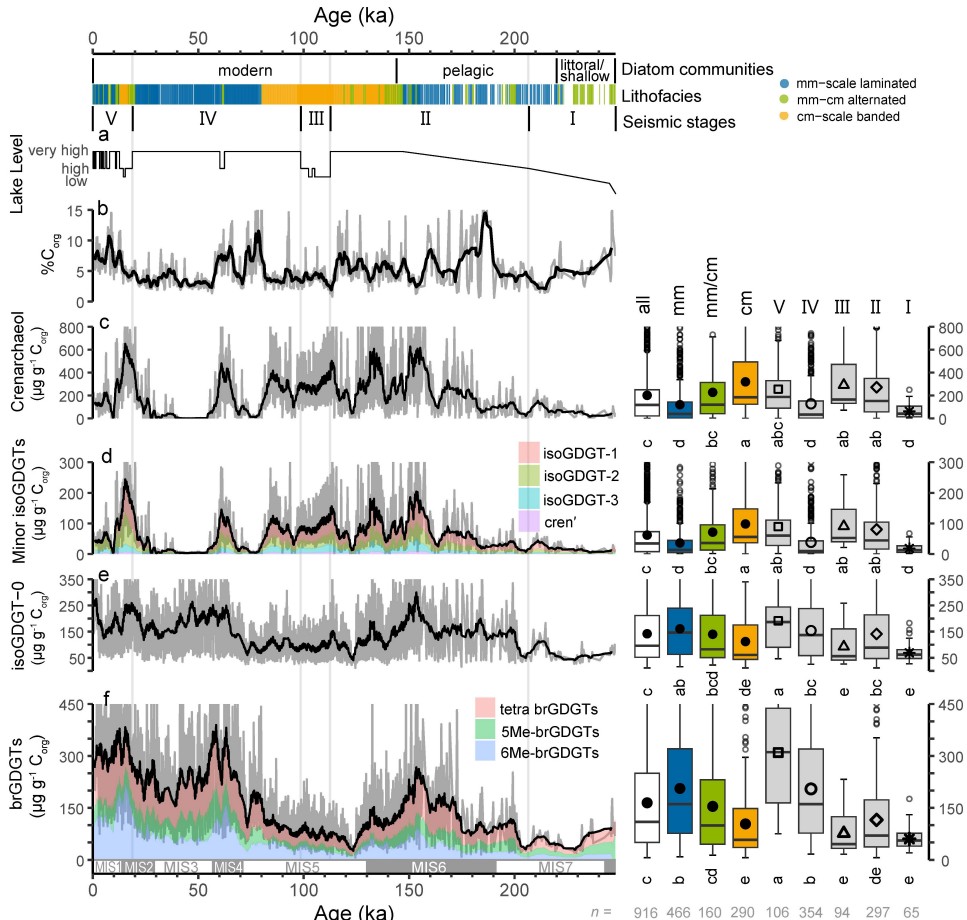

**Figure 3.** Down-core profiles of the organic-specific concentration of selected (groups of) GDGTs in the DeepCHALLA sediment sequence, in relation to Chala lake-system evolution over the past 250 kyr. Indicated on top is the timing of three major phases in the diatom communities in the DeepCHALLA sequence (after Tanttu, 2021), the lithofacies category of each sediment horizon (colored bar), and the depositional stages (V-I) based on the seismic stratigraphy of Lake Chala (Maitituerdi et al., 2022). Subsequent panels show (a) the lake level reconstruction based on the seismic reflection data (Maitituerdi et al., 2022), (b) the percentage of organic carbon (%TOC) content, and TOC-normalized concentrations of (c) crenarchaeol, (d) the minor isoGDGTs (isoGDGT-1, -2, -3, and the crenarchaeol isomer), (e) isoGDGT-0, and (d) the brGDGTs. Black curves represent the 10-point rolling averages of the full data series, shown in grey. The boxplots associated with the concentration profiles are grouped by the lithofacies category (using the same colour legend), and by depositional stage. The boxplots indicate the median (black line) and average (solid or open shape) values, and the first and third quantiles (lower and upper hinges). Whiskers (thin lines) extend to 1.5 × the interquartile range (IQR) from the hinges, with outliers (open black circles) defined as data points beyond this range. Letters in the box plots indicate statistically significant groups of data determined using Tukey multiple comparison of means with a 95% family-wise confidence interval; means which do not share letters are significantly different according at the 5% level of significance. Note that in both the depth profiles (b-f) and associated boxplots the y-axis has been stretched such that many high datapoints, and boxplot outliers fall off-scale, in order to improve readability of the 10-point rolling average trends. At the bottom the timing of the marine isotope stages (MIS) as defined by Lisiecki and Raymo (2005) is shown for reference.



## 4   Results

### 4.1   Trends in isoGDGT and brGDGT concentrations

The overall composition of GDGTs in the drilled Lake Chala sediments is dominated by crenarchaeol and isoGDGT-0, with mean fractional abundances of 0.30 and 0.28 respectively, followed by tetramethylated (0.14), 6-Me (0.10) and 5-Me brGDGTs (0.07). GDGT concentrations display a high degree of high-frequency variability with major swings between adjacent samples (Fig. 3). Smoothing of the time series (black curves indicate a 10-point rolling mean) reveals that substantial changes also occurred with regard to longer-term trends. The concentrations of crenarchaeol (Fig. 3c; range = 0–2740 $\mu$g g$^{-1}$ C$_{org}$, average

= 201 $\mu$g g$^{-1}$ C$_{org}$) and the less abundant isoGDGTs used to calculate TEX$_{86}$ ("minor" isoGDGTs) (Fig. 3d; range = 0.2–900 $\mu$g g$^{-1}$ C$_{org}$, average = 62 $\mu$g g$^{-1}$ C$_{org}$) are highly correlated (R = 0.99, p < 0.001; Fig. S3), as they both fluctuate in the same semi-regular pattern throughout the record with the exception of two longer periods withing Stages IV from c. 80–70 ka and c. 50–30 ka during which these compounds are often nearly absent. Concentrations of isoGDGT-0 (Fig. 3e; range = 11–740 $\mu$g g$^{-1}$ C$_{org}$; average = 142 g$^{-1}$ C$_{org}$) and the summed brGDGTs (Fig. 3f; range = 6–810 $\mu$g g$^{-1}$C$_{org}$; average

= 164 $\mu$g g$^{-1}$C$_{org}$) are also highly correlated with one another (R = 0.83, p < 0.001; Fig. S3). The concentrations of all GDGT are notably lower in the lowermost, oldest portion of the record compared to later stages (Fig. 3). For example, average summed concentrations of iso- and brGDGTs during depositional Stage I are respectively 149 and 60 $\mu$g g$^{-1}$ C$_{org}$: in both cases only 27% of their average concentrations in the full record (405 and 164 $\mu$g g$^{-1}$ C$_{org}$). All GDGT concentrations generally increase from the oldest horizon until a pronounced maximum dated to c. 153 ka (i.e., halfway through Stage II),

and notably display highly similar trends during this interval, which continues until c. 125 ka near the end of Stage II when all GDGT concentrations experience a pronounced minimum. Using the diatom-guided lake phases as reference, correlations between the concentrations of crenarchaeol, minor isoGDGTs (isoGDGT-1, -2, -3, and cren′), isoGDGT-0 and brGDGTs are universally quite high (R = 0.69–0.99, p < 0.001; Fig. S5) in the period preceding the appearance of the modern diatom community (c. 250–144 ka), whereas during c. 144–0 ka (Fig. S7) the concentrations of crenarchaeol and minor isoGDGTs are strongly correlated (R = 0.99, p <0.001), and likewise isoGDGT-0 and the brGDGTs (R = 0.84, p < 0.001) but correlations

across these two groups are severely reduced (R = 0.25–0.52) and not statistically significant (p > 0.05), as is also the case in the present-day lake (see section 2.3). The broad interval of c. 125–75 ka is characterized by lower-than-average isoGDGT-0 and brGDGT concentrations, after which concentrations increase again and generally remain high throughout the upper part of the record.

Samples extracted from mm-scale laminated or cm-scale banded sediments are characterized by significantly different GDGT concentrations (boxplots in Fig. 3). Mm-scale laminated sediments contain higher amounts of isoGDGT-0 (average = 146 $\mu$g g$^{-1}$ C$_{org}$) and brGDGTs (average = 206 $\mu$g g$^{-1}$ C$_{org}$), and lower amounts of crenarchaeol (average = 120 $\mu$g g$^{-1}$ C$_{org}$) and minor isoGDGTs (average = 36 $\mu$g g$^{-1}$ C$_{org}$) compared to cm-scale banded sediments (which have averages of respectively 112, 103, 319 and 98 $\mu$g g$^{-1}$ C$_{org}$, all differences being statistically significant at the 5% level of significance.

Sediments described as consisting of an alternation between these two lithofacies contain intermediate concentrations of these four classes of GDGTs, also being statistically different from these categories.





**Figure 4.** Principal component analyses (PCAs) of the fractional abundances of (a) all GDGTs, (b) the isoGDGTs, (c) brGDGTs, and (d) the minor isoGDGTs (i.e., isoGDGT-1, -2, -3 and the crenarchaeol isomer) in the DeepCHALLA sequence. The loadings of GDGTs is shown in the plot with arrows. In cases where GDGTs contributed < 1% of the variability of the data on both PC1 and PC2, they were removed to improve readability (but their fractional abundances were still included during analysis). Individual scores of DeepCHALLA sediment horizons are colored according to lithofacies, with seismic stage indicated by different symbols. Red numerals V-I represent the centroid values (i.e., average PC scores) of the sequence separated according to the depositional stages. Note that separate PC axes indicate the position of the individuals (sediment horizons) and variables (GDGT vectors) and that the amount of variability within the dataset predicted by PC1 and PC2 is provided as a percentage (in brackets).





## 4.2 Trends in the distribution of individual GDGTs

Principal component analysis (PCA) was performed on the fractional abundances of all 22 measured GDGTs together, and separately on the brGDGTs, isoGDGTs and minor isoGDGTs (Fig. 4). First considering the full suite of GDGTs, the first
principal component (PC1) accounts for 91.3% of the variance, and is chiefly related to the strongly opposing behavior of crenarchaeol versus isoGDGT-0 and the brGDGTs, in particular brGDGTs Ia, IIa′ and IIa (Fig. 4a). PC2 accounts for only 6.9% of the variance and mainly separates the isoGDGTs from the brGDGTs. In the PCA of only the isoGDGTs (Fig. 4b), PC1 accounts for a remarkable 99.7% of the variance, and as seen in the PCA on the full suite of GDGTs, relates overwhelmingly to the opposing behavior of crenarchaeol and isoGDGT-0. PC2 accounts for only 0.2% of the variance, and mainly separates
isoGDGT-1 from isoGDGT-2 and -3. In the PCA of only the minor isoGDGTs contributing to the $TEX_{86}$ proxy (i.e., isoGDGT-1, -2, -3, and cren′; Fig. 4d), PC1 accounts for 74.3% of the variance and is mainly controlled by the differing loadings of isoGDGT-1 and isoGDGT-2. PC2 accounts for 24.3% of the variance, with isoGDGT-3 plotting strongly on the negative side along with the weak negative position of cren′, versus the positive position of isoGDGT-1 and -3. Finally, in the PCA highlighting the differing distributions of individual brGDGTs (Fig. 4c), PC1 accounts for 65.1% of the variance, mainly
separating the 6-Me brGDGTs (in order of contribution: IIa′, IIIa′ and IIb′), which plot on the positive side, from the acyclic tetramethylated brGDGT Ia and the pentamethylated 5-Me brGDGT IIa, which plot on the negative side. PC2 accounts for 15.4% of the variance, with as largest contributors the opposing groups of IIIa′ and IIa versus Ia and IIb′.

In all PCAs described above there are noticeable differences between the PC scores of samples originating from each of the three lithofacies as defined above (colour codes in Fig. 4). Generally, samples from cm-scale banded sediments show greater
contributions of crenarchaeol, the crenarchaeol isomer, 6-Me brGDGTs and isoGDGT-3, whereas samples from mm-scale laminated sediments have a greater contribution of isoGDGT-0 and the 5-Me brGDGTs. In the PCA of the brGDGTs, ~30 samples from cm-scale banded and cm-mm alternating sediments deposited during Stage V are notably separated from all other samples (Fig. 4c; upper right hand corner) by their unusually high fractional abundances of IIIa′.

## 4.3 Trends in GDGT-based proxies

### 545 4.3.1 The BIT index

As presented above, PCA on the full suite of GDGTs strongly separates the brGDGTs from crenarchaeol along PC1. Consequently, variation in the BIT index (Fig. 5b) is highly correlated (R = 0.99, p < 0.001) to the PC1 scores of the respective sediment horizons in this PCA (Fig. 6a; Fig. S3). This result demonstrates that this long-defined proxy (Hopmans et al., 2004) captures the most significant variability in GDGT distributions throughout the DeepCHALLA sequence. Lake Chala sediments
cover nearly the full range of possible BIT-index values (range = 0.06–1; Fig S2) with an overall average value of 0.5. From the oldest sediments up to c. 160 ka, variation in the BIT index is erratic, switching rapidly between high (> 0.8) and low (< 0.3) values without discernable long-term trends. From c. 160 ka onwards changes in the BIT index become less erratic, and from c. 138 ka to c. 84 ka sustained periods of very low values (< 0.2) are interrupted by periods with predominantly high BIT values. In particular, the BIT index is consistently low (< 0.2) during c. 114–96 ka, largely overlapping with Stage III which



**Figure 5.** Down-core profiles of selected GDGT-derived proxies in the DeepCHALLA sediment sequence, in relation to Chala lake-system evolution over the past 250 kyr. Indicated from top to bottom, the timing of three major phases in the diatom communities in the DeepCHALLA sequence (Tanttu, 2021), the lithofacies category of each sediment horizon (colored bar), and the depositional stages (V-I) based on the seismic stratigraphy of Lake Chala, as well as (a) the lake level reconstruction based on the seismic reflection data (Maitituerdi et al., 2022). GDGT-based ratios and proxies from the DeepCHALLA sediment sequence are shown: (b) The BIT Index, (c) isoGDGT-0/crenarchaeol ratio, (d) f[CREN′], (e) %isoGDGT-2, (f) degree of cyclisation (DC), and (g) $IR_{6Me}$. Red points indicate values for which BIT index > 0.8, isoGDGT-0/cren. > 0.7, f[CREN′] > 0.04, and %isoGDGT-2 > 45% (see methods). The associated boxplots are as described in Fig. 3. Also shown is the timing of the marine isotope stages (MIS) as defined by Lisiecki and Raymo (2005).





is identified as a pronounced lake low-stand episode based on seismic reflection data (Maitituerdi et al. 2022; Fig. 5a-b). The lowest BIT index values of the entire record (0.06) occur at the very start of this interval. From c. 80 ka onward, corresponding almost exactly with a sustained lithofacies shift from cm-scale banded to mm-scale laminated sediments, a major change is observed in the nature of BIT-index variability, with values approaching unity that are sustained for longer periods. BIT-index values are continuously high until c. 24 ka (roughly corresponding to the Stage IV to V transition), except for a ~10-kyr long episode c. 62-52 ka and a brief interruption dated to c. 37 ka. Sustained low BIT-index values between 20 ka and 14 ka again correspond to a sustained period of cm-scale banded sedimentation interpreted as a pronounced lake low-stand (Maitituerdi et al., 2022). Thereafter, BIT-index values rise steadily before experiencing a brief reversal to low BIT values 13–11 ka, followed by a period with values close to unity during 11–9 ka. Over the last 6 kyr, the Chala BIT index has fluctuated around 0.6.

Comparison of the BIT index time series with those of the absolute concentrations of crenarchaeol and the summed brGDGTs reveal that variation in the BIT index is predominantly influenced by the abundance of crenarchaeol and considerably less by the contribution of brGDGTs (Fig. 7a–b), in agreement with settling particle data (Baxter et al., 2021). However, sediment horizons with BIT-index values < 0.2 generally have brGDGT concentrations < 300 $\mu$g g$^{-1}$ C$_{org}$, suggesting that brGDGT variability may also affect BIT-index variation at least to a small extent. Once again, the average BIT index of the three lithofacies categories differs significantly (boxplots in Fig. 5b), with cm-scale banded sediments being characterized by low values (average = 0.2, including a handful of distinct outliers), whereas mm-scale laminated sediments generally have much higher values (average = 0.7), and samples from the mm-cm scale alternating lithofacies most often display intermediate values (average = 0.4).

### 4.3.2 IsoGDGT-derived proxies

Three additional proxies based on ratios between isoGDGTs that have been used to investigate changes in Thaumarchaeotal production (Sinninghe Damsté et al., 2012a; Baxter et al., 2021) are the isoGDGT-0/cren ratio, f[CREN′], and %isoGDGT-2 (Fig. 5c–e). The isoGDGT-0/cren ratio is strongly correlated to PC1 of the PCA on all GDGTs (R= 0.59, p < 0.001; $\rho$ = 0.99, p < 0.001; Fig. S3) and PC1 of the isoGDGTs ($\rho$ = 1.0, p < 0.001; Fig. 6d; Fig. S3). Hence also the isoGDGT-0/cren ratio and the BIT index are highly correlated (R= 0.61, p < 0.001; $\rho$ = 0.95, p < 0.001; Fig. 7f), as expected owing to both of them being controlled by crenarchaeol abundance, and the overlapping ecological niches of brGDGTs and isoGDGT-0 in the anoxic lower water column of Lake Chala (Baxter et al., 2021). Variability in the isoGDGT-0/cren ratio is highly erratic in the lower part of the DeepCHALLA sequence (c. 250–160 ka) and becomes more structured thereafter, mirroring the trends displayed by the BIT index. f[CREN′] is moderately correlated with PC1 scores of all GDGTs (R = 0.51, p < 0.001) and of the isoGDGTs separately (R = 0.58, p < 0.001), and more strongly correlated to isoGDGT-0/cren (R = 0.70, p < 0.001; Fig. 7g; Fig. S3). f[CREN′] attains considerably low values (< 0.025) in most of the DeepCHALLA sequence (Fig. 5d), presumably indicating an often limited presence of Group I.1b Thaumarchaeota (Baxter et al., 2021). The very few instances of higher f[CREN′] values before 60 ka mostly consist of single sediment horizons. Sustained periods of high f[CREN′] values (> 0.04) occurred from c. 60 ka to 25 ka, mostly in mm-scale laminated sediments deposited during the middle and latter part of Stage IV, and



from 11 ka to 9 ka, also in mm-scale laminated sediments. The %isoGDGT-2 proxy (Fig. 5e) is strongly inversely correlated to
PC1 of the minor isoGDGTs (R = -0.93, p < 0.001). Before c. 80 ka its value hovers remarkably stably around 30%, with only
two instances of more elevated values. After this time its baseline shifts to higher percentages. From c. 55 to 24 ka within Stage
IV, %isoGDGT-2 is highly variable, and frequently exceeds 50%. Two other periods of generally high %isoGDGT-2 values
occurred between 20–11 ka and 9–0 ka.

$TEX_{86}$ is strongly correlated to PC1 of the minor isoGDGTs (R = -0.95, p < 0.001; Figs. 6e and S3). From the oldest
sediment horizon until c. 180 ka, the index is relatively stable around 0.45, after which it increases to ∼0.55 by c. 165 ka,
remaining at that level until c. 80 ka (Fig. 8b). The period c. 80–20 ka is characterized by highly variable $TEX_{86}$ values
ranging between 0.43 and 0.77. At 20 ka the index rises dramatically, reaching peak values of 0.85 at ∼5 ka and remaining
high until the top of the sequence.

Average isoGDGT-0/cren ratio, f[CREN′], and %isoGDGT-2 values of mm-scale laminated and cm-scale banded sediments
differ significantly (boxplots in Figs. 5c–d), with mm-scale laminated sediments having higher average values (39.6, 0.04, and
37.1%) than cm-scaled banded sediments (1.1, 0.01, and 30.2%) for all three proxies, values in mm-cm scale alternating sed-
iments being intermediate (isoGDGT-0/cren ratio, %isoGDGT-2) or near-identical to cm-scale banded sediments (f[CREN′]).
In contrast, average $TEX_{86}$ values of the three lithofacies are not significantly different from one another (boxplots in Fig. 8b).

### 4.3.3 BrGDGT-derived proxies

The $IR_{6Me}$ of penta- and hexa-methylated brGDGTs varies substantially throughout the DeepCHALLA sequence (0.24-0.89,
average = 0.61) (Fig. 5g, note the reversed y-axis). There is strong correlation between $IR_{6Me}$ and PC1 of the PCA on brGDGTs
(R = 0.95, p <0.001; Fig. 6b), meaning that this ratio reflects an important aspect of variability in the distribution of brGDGTs
in Lake Chala sediments. $IR_{6Me}$ values are relatively low and stable in the early part of the time series, becoming more variable
from c. 170 ka onwards. Temporal variability in $IR_{6Me}$ shows some similarities to that of the BIT index (R = -0.49, p < 0.001;
Fig. 7c) and isoGDGT-0/cren ratio (R = -0.33, p < 0.001) reflected in moderate negative correlations with those proxies over
the full time series. At c. 142 ka a sharp transition occurs from low to high $IR_{6Me}$ values, almost coeval with the sustained
transition to cm-scaled banded sedimentation at that level and establishment of the modern-day diatom community. From this
time onwards similarity between trends in $IR_{6Me}$ and BIT index (and also isoGDGT-0/cren) is enhanced, resulting in stronger
negative correlation between $IR_{6Me}$ and the BIT index after 144 ka (R = -0.64, p < 0.001; Fig. S7). Comparing variation
in $IR_{6Me}$ to the concentrations of 5-Me and 6-Me brGDGTs indicates that both groups of brGDGTs have a comparable
influence on the $IR_{6Me}$ signal (Fig. 7d–e). For example, sediments with the highest $IR_{6Me}$ values (> 0.7) also have the highest
concentrations of 6-Me brGDGTs, while those with $IR_{6Me}$ below 0.7 generally have greater amounts of 5-Me brGDGTs.

The CBT′ index, typically used in pH reconstruction, is partly affected by the degree of cyclisation of the brGDGTs, but as
shown by its strong positive correlation with $IR_{6Me}$ throughout the DeepCHALLA sequence (R = 0.95, p < 0.001; Fig. S3), in
Lake Chala this index is mainly controlled by variation in the relative abundance of 5-Me and 6-Me brGDGTs. Accordingly,
the CBT′ index also shows strong correlation with PC1 of the PCA on the brGDGTs, which separates the 5-Me and 6-Me
brGDGTs (R = 0.99, p < 0.001; Fig. S3). On the other hand, the degree of cyclisation of the brGDGTs (expressed with the DC





**Figure 6.** Scatter plots comparing PC scores and GDGT ratios. (a) PC1 scores from PCA of all the GDGTs compared to the BIT index. From PCA of the brGDGTs, (b) PC1 scores are compared to $IR_{6Me}$ and (c) PC2 scores are compared to $MBT'_{5Me}$. (d) PC1 scores from PCA of the isoGDGTs compared to the isoGDGT-0/crenarchaeol ratio. (e) PC2 scores from PCA of the minor isoGDGTs (isoGDGT-1 to -3, and cren′) compared to $TEX_{86}$. Data points are colored according to lithofacies and the depositional stage is indicated by the point shape. Red numerals (V-I) represent the average values of the sequence grouped according to the depositional stage.





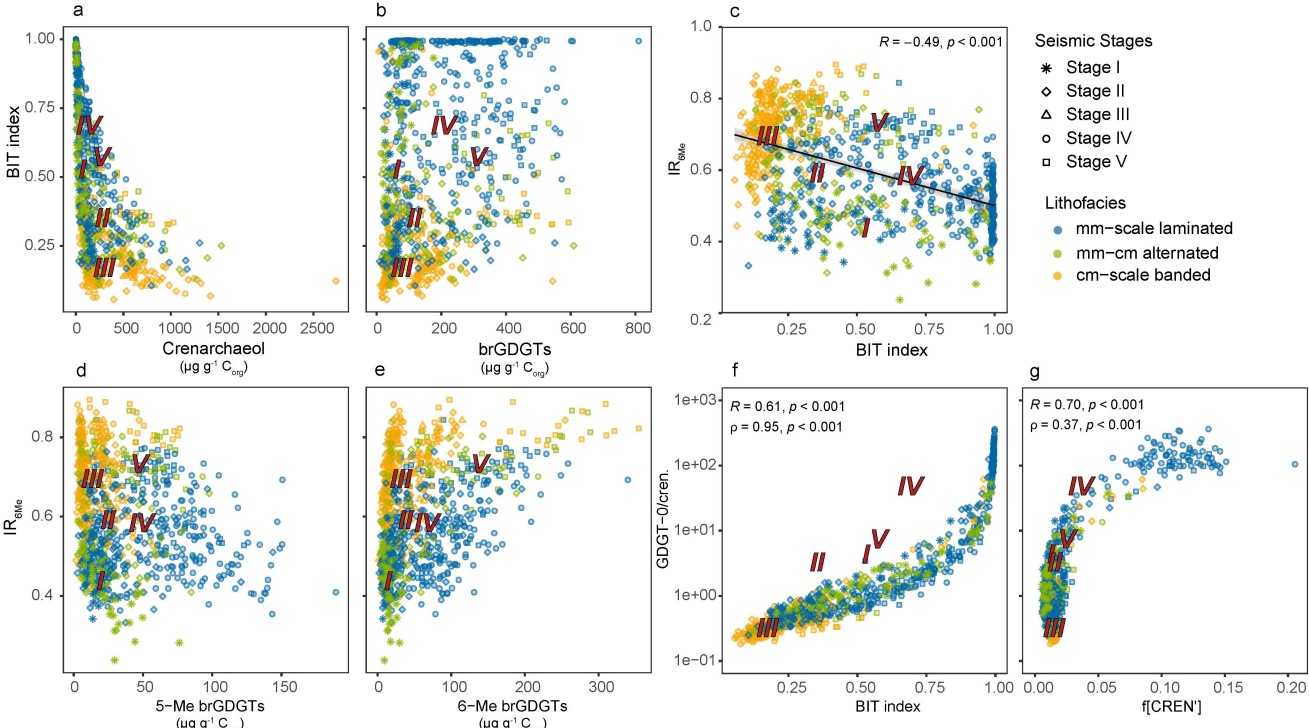

**Figure 7.** Scatter plots comparing the BIT index versus the concentration of (a) crenarchaeol and (b) brGDGTs, $IR_{6Me}$ versus (c) BIT index, and the summed concentration of (d) the 5-Me brGDGTs, and (e) the 6-Me brGDGTs, (f) the BIT index versus the isoGDGT-0/cren ratio, and (g) f[CREN′] versus the isoGDGT-0/cren ratio. The sediment horizons are colored according to lithofacies, and the depositional stage is indicated by the point shape. Red numerals (V-I) represent the average values of the sequence grouped according to the depositional stages.

ratio) shows only a weak positive correlation with $IR_{6Me}$ (R = 0.35, p < 0.001; Fig. S3), and is also only weakly correlated with either PC1 and PC2 of brGDGT distribution (R = 0.31, p < 0.001; R = -0.30, p < 0.001). This suggests that DC is not a strong measure of the variance in the distribution of brGDGTs in Lake Chala sediments . Accordingly, its time series does not show large changes, remaining mostly around 0.2, except during the period between c. 140 ka and c. 100 ka and in the last 10 kyr when larger temporal variability is observed (Fig. 5f).

$MBT'_{5Me}$ shows strong correlations to both PC1 (R = 0.66, p < 0.001; Fig. 6c; Fig. S3) and PC2 (R = -0.68, p < 0.001) of the brGDGTs, and hence, naturally also to $IR_{6Me}$ (R = 0.84, p < 0.001) and CBT′ (R = 0.68, p < 0.001). This index ranges from 0.48–0.85, and besides a sharp rise at the base of the record, is relatively stable in sediments deposited before c. 175 ka (Fig. 8c). Thereafter, $MBT'_{5Me}$ first increases to near-maximum values (∼0.8) at c. 160 ka and then drops abruptly to a sustained minimum (∼0.6) dated to c. 150–140 ka which terminates as sharply as it started. The period from c. 140 ka until c. 80 ka is characterized by variable but generally high $MBT'_{5Me}$ values (frequently above 0.8), again ending in an abrupt drop to values



**Figure 8.** Down-core profiles of select GDGT-based paleotemperature proxies in the DeepCHALLA sediment sequence, in relation to Chala lake-system evolution over the past 250 kyr. Indicated from top to bottom, the timing of three major phases in the diatom communities in the DeepCHALLA sequence (Tanttu, 2021), the lithofacies category of each sediment horizon (colored bar), and the depositional stages (V-I) based on the seismic stratigraphy of Lake Chala, as well as (a) the lake level reconstruction based on the seismic reflection data (Maitituerdi et al., 2022). (b) $TEX_{86}$ values for the DeepCHALLA sediment sequence with red circles reflecting samples which are excluded according to the filtering method of Baxter et al. (2021). Black line connects those sediment horizons which remain after the filtering. (c) $MBT'_{5Me}$ index and (d) mean daily insolation for the months of September (blue line) and March (grey line) at the equator. (e) Mean Summer Temperature (MST) calculated according to Pearson et al. (2011), rescaled using an ensemble reconstruction of temperature records from seven other East African lakes following Baxter et al. (2023). The associated boxplots are represented as described in Fig. 3. Also shown is the timing of the marine isotope stages (MIS) as defined by Lisiecki and Raymo (2005).




below 0.7. $\text{MBT}'_{5\text{Me}}$ reaches its lowest values just after c. 60 ka, then gradually increases until 14 ka when a short-lived spike

occurs, lasting until 11 ka and followed by a gently rising then falling trend over the last 10 kyr.

The time series of MST reconstructed using the global lake calibration of Pearson et al. (2011) (see Methods) spanning the complete DeepCHALLA sequence is strongly correlated to PC2 of the brGDGTs (R = -0.65, p < 0.001, Fig. S3) and to the DC (R = 0.76, p < 0.001). Rescaled MST values (Baxter et al., 2023) range between 20.3 °C and 25.6 °C. As reconstructed, rescaled MST is rather stable (generally 22–23 °C) from c. 250 ka until 180 ka, at which time it gently increases to ∼24 °C then again

decreases to a pronounced minimum (∼21.5 °C) dated to c. 145 ka (Fig. 8e). After this, MST increases rapidly, reaching peak values at c. 134 ka. Following another minimum bottoming out c. 126 ka, another period of frequently above-average MST occurs at c. 121–108 ka. From c. 108 to 24 ka, rescaled MST generally hovers between 22 and 23.5 °C, with the exception of an inferred cooler episode (∼21.5 °C) centered at c. 60 ka. Sustained low MST values also occur from 21 to 14 ka. Thereafter, MST rises sharply and the period 8.5–4.5 ka is inferred to have been marked by higher than average temperatures, occasionally

reaching ∼25 °C. In sediments deposited during the last 4 kyr, MST is mostly between 23 and 24 °C.

As is the case with most GDGTs, also average values of the proxies $\text{IR}_{6\text{Me}}$, DC, $\text{MBT}'_{5\text{Me}}$ and rescaled MST (°C) are significantly different between samples extracted from mm-scale laminated and cm-scale banded sediments, now with cm-scale banded sediments producing higher values on average (respectively 0.71, 0.30, 0.72 and 23.0; see boxplots in Figs. 5 and 7) than mm-scale laminated sediments (0.54, 0.32, 0.66 and 22.7).

**4.4 Periodicities in GDGT concentrations and proxies**

Periodicity analysis performed on selected down-core profiles of GDGT concentrations and proxies (Fig. 9; Figs. S9–S11) revealed strong cyclicities likely related to orbital precession (23 kyr) and perhaps obliquity (41 kyr). For example, periodicities of 22.2 and 44.4 kyr occur in the concentration of crenarchaeol, which become apparent from respectively c. 180 ka and c. 110 ka onwards, respectively (see wavelet analysis; Fig. 9c). The absence (or weakness) of these cycles in the older portion of

the crenarchaeol time series is also evident in the varying amplitude of band-pass filters with those periodicities (Fig. 9b): the amplitude of the precession- and obliquity-period filters is very modest in the oldest sediments and increases steadily in the course of lake-basin Stages I and II. Moreover, in the global wavelet spectrum restricted to the period 180–0 ka, the signatures of precession- and obliquity-scale cycles are enhanced compared to the wavelet spectrum covering the complete time series, and the cycle lengths shift closer to the astronomical solutions (Figs. 9d and 9e). Also the time series of the BIT index and

$\text{IR}_{6\text{Me}}$ display periodicities close to the 23-kyr and 41-kyr cycles of precession and obliquity (Figs. S9–S10. Likewise, although less clear as in the case of crenarchaeol, also the wavelet spectra and band-pass filtered time series of these two proxies indicate weak expression of the astronomical cycles during Stage I and the first half of Stage II, becoming more pronounced when analysis is restricted to the last 180 kyr.



**Figure 9.** Periodicity analysis of the concentration of crenarchaeol. Indicated from top to bottom, the timing of three major phases in the diatom communities in the DeepCHALLA sequence (Tanttu, 2021), the lithofacies category of each sediment horizon (colored bar), and the depositional stages (V-I) based on the seismic stratigraphy of Lake Chala, as well as (a) the lake level reconstruction based on the seismic reflection data (Maitituerdi et al., 2022). (b) The concentration of crenarchaeol in light grey with green and pink curves representing the band pass filtered periods reflected by wavelet analysis which are comparable to the periods of precession and obliquity, respectively. (c) Wavelet analysis using morlet function, with the corresponding (d) wavelet spectrum resulting from analysis of the full sequence DeepCHALLA sequence. Warm and cold colors reflect high and low values of the power spectrum, respectively. (e) Wavelet spectrum from spectral analysis of the sedimentary sequence representing 180–0 ka only, showing the more pronounced precession and obliquity cycles. The red stippled line in (d) and (e) represents the 95% confidence interval; the dominant frequencies are labelled at the top of the maxima. Also shown is the timing of the marine isotope stages (MIS) as defined by Lisiecki and Raymo (2005).



## 5 Discussion

### 665    5.1    Influence of lake basin evolution on GDGT niches

The low concentrations of all GDGTs during the first c. 50–70 kyr of lacustrine deposition in the Lake Chala basin (Fig. 3) could suggest poorer preservation of GDGTs due to longer exposure to oxygen while settling through the water column and/or in the surface sediments prior to permanent burial, or their apparent dilution because of greater amounts of organic carbon. However, GDGTs are considered to be resilient to degradation at least on the timescale of this study, and found abundantly in

well-oxygenated environments including aerated soils (Schouten et al., 2013). Moreover, the sedimentary $C_{org}$ content during this early period is comparable to that of later stages (Fig. 3b). Therefore, we interpret the low concentrations of all GDGTs during the first c. 50–70 kyr (Fig. 3) to reflect the prevalence of a shallower and/or better mixed water column than today during Stage I and the early part of Stage II, which was less favorable to either brGDGT producers or Thaumarcheota due to lack of well-developed anoxic and sub-oxic zones (Fig. 10a). In agreement with the low GDGT concentrations, seismic analysis also

suggests that lake level was markedly lower than today during Stage I (c. 248–207 ka) (Maitituerdi et al., 2022), and also the high relative abundance of benthic diatoms in first c. 30 kyr of the DeepCHALLA sequence confirm the proximity of shallow-water habitat (Tanttu, 2021). The disappearance of benthic diatoms c. 220 ka (Tanttu, 2021) suggests that water level rose strongly from at least this period onwards, and is concurrent with the first meaningful rise in GDGT concentrations. The first prolonged deposition of mm-scale laminated (varve-like) sediments around this time (Fig. 3) suggests that this deepening

of the water column also promoted greater persistence of bottom water anoxia. The sharp increase in GDGT concentrations at c. 210 kyr may suggest a rise in lake level, and corresponds to the boundary between Stage I and II (Maitituerdi et al., 2022).

Besides the lower-than-average GDGT concentrations before c. 200-180 ka, several other aspects of GDGT variability were different during this early depositional phase from later on. First, the GDGT concentrations (crenarchaeol, isoGDGT-0, minor isoGDGTs, and summed brGDGTs) universally show moderate to strong positive correlations during this period (e.g.,

250–180 ka; Fig. S4), contradicting our current understanding of their niche partitioning in the modern system, which predicts distinct behavior of the two groups of GDGTs associated with either the upper mixed layer or the anoxic zones (respectively, crenarchaeol and minor isoGDGTs versus isoGDGT-0 and brGDGTs). Correlations among these GDGTs reflect the known associations of these biomarkers with the different water column zones (for example, isoGDGT-0 and the brGDGTs being strongly correlated to one another but not to crenarchaeol) only when they are limited to the last c. 144 ka, i.e., the period during

which the diatom assemblages attest that the water column structure (and hence, dominant mode of nutrient recycling) in Lake Chala has been comparable to the present-day situation. Similar temporal differences can be seen in the correlations between the GDGT-derived proxies. For example, the inverse correlation between the BIT index and $IR_{6Me}$ improves from R = -0.31 (p < 0.001) in the section before c. 144 ka to R = -0.64 (p < 0.001) after c. 144 ka, reflecting the stronger connection between these proxies based on our understanding of GDGT niches in the modern system. Secondly, the variation in GDGT proxies associated

with water-column mixing and stratification (e.g., the BIT index, isoGDGT-0/cren) is highly erratic before c. 170 ka, while other proxies are seemingly unresponsive before c. 200 ka (e.g., $IR_{6Me}$, and f[CREN′]). Hence, at face value these proxies do not paint a cohesive history of changes in lake depth and mixing regime during this period. During Stage II (c. 207–113





**Figure 10.** Schematic representation of our current understanding of the development of Lake Chala (both water column and sediment infill) over the last 250 ka based on inferences from settling particle and sediment trap data (Sinninghe Damsté et al., 2009; Buckles et al., 2014; van Bree et al., 2020; Baxter et al., 2021) and seismic profiling of the Chala basin (Maitituerdi et al., 2022). The ecological niches and distribution of GDGTs and their producers are shown for the five major depositional stages. Lake bottom sediments are shown in grey shading.





ka), seismic stratigraphy reveals a gradual transition from predominantly ponded to draped sedimentation, indicating that total water depth increased steadily, meaning that throughout this period the surface level of Lake Chala must have risen faster than the rate of sediment accumulation. However, due to the near-vertically sloping crater walls this lake deepening resulted in only a very modest increase in lake surface area (Maitituerdi et al., 2022). Comparative morphometric analysis and water-column profiling in 60 volcanic crater lakes in western Uganda found that the depth of water-column mixing in these systems is most strongly related to lake surface area, not the relative height of the crater rim thought to provide wind shelter (De Crop and Verschuren, 2021). This means that in Lake Chala, where lake surface area is more or less constant across a large range of lake depths (Maitituerdi et al., 2022), changes in lake surface level alone cannot be responsible for changes in the depth of the mixed layer. In the absence of >100 m of sediments deposited later on, the lake basin during Stage II was substantially deeper than today, and likely the water column of Lake Chala attained its overall maximum height (>200 m) during this stage. Under such conditions, mixing of the entire water column might be expected to have been much less likely than today. However, given the thinner sediment infill at this time, the crater bottom and lower side walls were probably still relatively porous, allowing greater subsurface outflow and thus removal of any dissolved solids that might otherwise accumulate in the lower water column by decomposition of organic matter, or following the dissolution of photosynthetically precipitated calcite. This more 'leaky' nature of the crater basin would have prevented development of the endogenic chemical density gradient that might promote permanent stratification (biogenic meromixis sensu; Hutchinson 1937), and thus allowed at least occasional mixing of the deeper water column at multi-annual time scales, despite great lake depth. Lacking chemical stratification, it was more likely that under the right conditions (for example, exceptional lake-surface cooling during an extremely windy dry-season episode) this deep water column could fully turn over, allowing oxygen to penetrate to the water-sediment interface. As such, not only Stage I but also the deep early phase of Stage II were likely characterized by sporadic short-lived events of complete mixing, contrasting with the present-day lake state of permanently stratified, and permanently anoxic, lower water column. We suggest that the highly erratic behaviour of GDGT-based stratification proxies such as the BIT index, isoGDGT-0/crenarchaeol, and f[CREN′] prior to c. 170 ka may be explained by the occasional occurrence of these deep-mixing events, with intermittent peak values indicating that episodes of multi-annual stratification did occur during this interval, but inconsistently (Fig. 5). In line with this, although the mostly mm-scale laminated sediments deposited during Stages II imply that bottom-water anoxia prevailed, their frequent interruption by alternated mm-cm scale lamination confirms that erratic events of complete mixing did occur. Most importantly, lack of chemical stratification in the deep-lake environment of Stage II meant that meromictic conditions were maintained (at least most of the time) by the temperature gradient alone (thermogenic meromixis sensu; Katsev et al. 2010), and thus that the water column of Lake Chala was not separated into the six well-defined depth zones as it is today (Buckles et al. 2014; Fig. 10a). Specifically, the equivalent to Zones 4–5 as recognized in the modern system, which presently form the permanent anoxic zone and are characterized by higher conductivity and lower pH compared to the upper mixed layers (Zones 1–3), most probably did not yet exist (Fig. 10b). Hence the depth of oxygen penetration during seasonal deep-mixing events must have been highly variable as it was not restricted by a static chemocline (De Crop and Verschuren, 2021). Moreover, due to the more dilute water mass and the lower primary productivity typically associated with deep lakes, perhaps also the visible light and UV penetration depths differed from today (Secchi disk depth seasonally varying between





1 and 8 m; van Bree et al. 2018; Fig. 10a–b). As Thaumarchaeota are known to be photosensitive (Merbt et al., 2012; Horak et al., 2018)) this may also have influenced their ability to grow in the upper water column and, hence, their presence and/or

maximum production. Given these marked differences in the ambient aquatic environment, it therefore seems unsurprising that relationships between different GDGTs and the derived proxies in this time interval do not mirror those observed in the modern system. As the association of the different GDGTs with particular niches in the water column underpins the understanding of climate-proxy relationships, simple extrapolation of this understanding to the earliest phases of lake basin evolution is not valid.

Further support for a diminished influence of climate on the GDGT proxies measured in the older sediments can be gleaned from the periodicity analysis. Namely, the apparent influence of orbital insolation forcing on certain GDGT concentrations and proxies (i.e., crenarchaeol concentration, BIT index, and $IR_{6Me}$) is strong in the younger portion of the time series but weak before c. 180 ka in the case of precession and before as recently as c. 90 ka in the case of obliquity (Fig. 9, Figs. S9-S10). Precession is known to strongly influence the monsoon system and terrestrial hydroclimate in tropical Africa, as it controls

the amount and distribution of solar radiation received there (Singarayer and Burrough, 2015). Due to varying eccentricity modulating the amplitude of precessional insolation forcing the strength of the precession signature in tropical African climate history can be expected to vary through time (Blome et al., 2012). However the amplitude of precession was markedly larger during c. 250-180 ka (late MIS8 to early MIS6) than during the last glacial period (MIS4–MIS2) (Fig. 8d). Delayed expression of the obliquity cycle relative to precession may partly relate to the growth of the polar ice sheets during the last glacial period

(MIS4–MIS2), enhancing the influence of high-latitude climate dynamics on the tropics (Tjallingii et al., 2008). However, this does not explain the lack of an obliquity signature during the penultimate glacial period (MIS6). Moreover the timing of when this cyclicity appeared is not consistent among proxies and in most cases began before the start of MIS4 (Fig. 9, Figs. S9-S10). Regardless, the absence of a strong signature of either obliquity or precession from the start of the DeepCHALLA record confirms that climate variablity was probably not the dominant mechanism driving variability in sedimentary GDGT

distributions during this period, unlike during the more recent lake basin stages.

Continuous accumulation of fine-grained sediments in the deep crater basin of Lake Chala (in total ∼70 m by c. 180 ka; Maitituerdi et al. 2022; Fig. 10b) gradually diminished the leaky nature of the basin floor. As a result, more dissolved solids were retained, in turn promoting the development of chemical density stratification supporting biogenic meromixis. Thus the distinct mixing Zones 4 and 5 probably developed during the later portion of Stage II (not shown in Fig. 10b). Based on our GDGT data,

this first time the water column of Lake Chala attained its modern-day structure of six mixing zones occurred c. 180-170 ka. Then around 144 ka the fossil diatom assemblages indicate a dramatic change in lake conditions and internal nutrient cycling (Tanttu, 2021). Both dominant diatom species in Lake Chala must be well-adapted to the nutrient-limited environment that is typical of deep lakes with small catchments, but as the more heavily silicified *A. barkeri* presumably requires more nutrients for population persistence than *Nitzschia* species (Tanttu, 2021), appearance of the former taxon in the DeepCHALLA sequence

c. 144 kyr ago testifies to improved upcycling of nutrients from the hypolimnion. Notably, the coincident transition to mainly cm-scale banded sedimentation at c. 145 ka (Fig. 3a) indicates that deep mixing at least occasionally reached the lake bottom causing modest sediment disturbance. This would have been promoted if a trend towards drier climatic conditions reduced





lake depth by lowering its surface elevation. However, as clear evidence of such an event is lacking in the seismic stratigraphy (Fig. 3a), this facies transition may be explained solely by shallowing of the water column due to the progressive infilling of sediments (∼100 m by that time; Maitituerdi et al. 2022). Together the evidence suggests that despite development of the six-zone 'modern' water column structure, the magnitude of chemical stratification at this time was still relatively modest, such that it could be overcome by the lake-surface cooling occurring during an exceptionally cool or windy deep-mixing season. During Stage III, a period of strongly basin-focused sedimentation implying a major drop in lake surface level (Moernaut et al., 2010; Maitituerdi et al., 2022), the reduction in lake volume due to prolonged negative water budget (lake-surface evaporation exceeding the sum of catchment precipitation and sub-surface inflow) increased the overall dissolved-ion concentration such that the resistance of chemical stratification to mixing approached conditions similar to today (Fig. 10c). Presumably, the transition to a more positive water budget ending this low-stand period c. 99 ka also acted to strengthen chemical stratification across the Zone 3/4 boundary by adding dilute water to the at least annually mixed surface layer (Zones 1-3), such that by c. 80 ka Lake Chala enjoyed stable meromixis, as evidenced by the almost uninterrupted deposition of mm-scale laminated sediments throughout the remainder of Stage IV. Hence, the sequence of events which first created the different depth zones in the water column and then enhanced their distinctness in terms of ambient physical and chemical conditions, allowed the niches of GDGT producers to become increasingly relatable to those observed in the modern-day lake system. As such, between c. 180 ka and c. 80 ka there is a progressively increasing consistency between lake conditions as inferred from the GDGT concentrations and indices based on modern-system understanding, and the lithofacies and seismic reflection data which reflect the long-term history of changes in respectively water-column mixing and lake depth.

From c. 140 to c. 80 ka, a period when lithostratigraphy indicates meromixis to have been less stable than before and after, there were several prolonged periods of low BIT-index values, low isoGDGT-0/cren ratios, and high $IR_{6Me}$ values, suggesting a relatively thicker oxygenated layer (Zones 1–2) and greatly reduced or absent anoxic lower mixed layer (Zone 3). This is also indicated by lower-than-average concentrations of the anoxically produced brGDGTs and isoGDGT-0. The combined evidence suggests that deep mixing was frequent but most often halted by the chemocline at the top of Zone 4, which in turn suggests that also total lake depth was lower during this broad period than before and after. In particular, there is near-perfect agreement between the longest period of consistently low BIT-index and isoGDGT-0/cren values c. 113–99 ka with the timing of the major Stage III lowstand evidenced by seismic stratigraphy (Fig. 5; Fig. 10d). Essentially all GDGTs and indices point to an unprecedented change in the structure of the water column c. 80 ka, when a loss of Thaumarchaetoa and increasing absolute and relative proportion of all GDGTs produced in anoxic waters suggest persistence of strongly stratified conditions, as also indicated by the equally abrupt lithofacies transition ifrom cm-scale banded to mm-scale laminated sediments. The seismic record shows that lake level was consistently high during the period c. 80-25 ka (the entire Stage IV; Fig. 10e), except for a brief interlude of ponded sedimentation (i.e., an inferred lake low-stand) centered at c. 60 ka, also registered as a brief interruption of mm-scale laminated sedimentation (Fig. 3a). During this interval (c. 80-25), the GDGT concentrations and proxies predominantly suggest an expanded anoxic zone under tall water column conditions and more limited upper water column mixing; reduced amounts of crenarchaeol and the minor isoGDGTs in comparison to isoGDGT-0 and the brGDGTs are reflected in the mostly high BIT-index and isoGDGT/cren values and further correspond generally low $IR_{6Me}$. Similarly





to the lithofacies and seismic profiling, the GDGTs and proxies responded to the brief lowstand at c. 60 ka. From 24 to 14 ka the GDGT proxies (e.g., BIT index, isoGDGT-0/cren and $IR_{6Me}$) are also consistent with the seismic and lithofacies data (Fig.

5) which suggest another period of reduced lake level and deep mixing (Fig. 10f), second in amplitude only to the Stage III lowstand. However it should be noted that by that time, the additional deposition of ~55 m of sediments had markedly reduced the magnitude of lake-level drop required to generate bottom disturbance, at a comparable position of the lake's surface level. After a period of very wet conditions between 12 ka and 9 ka indicated by the GDGTs (Fig. 10g), the ensuing period until the present is characterized by a moderately deep water column, with certain proxies (e.g., the BIT index) suggesting a somewhat

fluctuating lake level. The generally wet conditions suggested by the GDGT proxies are matched by a return to predominantly mm-scale laminated sedimentation.

### 5.2   Environmental control of the Lake Chala $IR_{6Me}$ record

In soils and peats (Weijers et al., 2007; Peterse et al., 2012; De Jonge et al., 2014; Xiao et al., 2015; Naafs et al., 2017a, b) strong correlations have been found between pH and the degree of cyclisation of brGDGTs (e.g., expressed in CBT′ index)

or the relative contribution of 6-Me brGDGTs (e.g., expressed in the $IR_{6Me}$). By contrast, although these relationships are sometimes reported for lake surface sediment datasets (e.g., Raberg et al. 2021; Dang et al. 2016), they are not unambiguous (e.g., Russell et al. 2018; Martínez-Sosa et al. 2021), possibly relating to the highly variable pH conditions with depth and season in lakes. In the DeepCHALLA sequence, it appears that the relative contribution of the 5-Me and 6-Me isomers has a strong influence on the CBT′ index and that the degree of cyclisation index (DC) is not an obviously important measure for

understanding variability in brGDGT distributions. Notably, whereas SPM data from Lake Chala (van Bree et al., 2020) do not clearly reveal the environmental significance of the DC and CBT′ indices, clear understanding emerged of $IR_{6Me}$ being a reflection of variation in the spatially distinct niches of 5-Me and 6-Me brGDGT producers in the water column (see section 2.5.3; van Bree et al. 2020). However the ambient pH range in their respective niches (~7.2–8 in Zone 3 and ~6.9–7.1 in Zones 4–5; Buckles et al. 2014; Baxter et al. 2021) opposes the trend found in soils, where the 6-Me isomers dominate in high

pH conditions (De Jonge et al., 2014). This ambiguity suggests that $IR_{6Me}$ variability in the Lake Chala sediment sequence is likely related to another factor than pH.

     Our mechanistic understanding of the modern system of Lake Chala strongly couples variation in the BIT index and $IR_{6Me}$ on the seasonal timescale and longer. An increase in $IR_{6Me}$ can be linked to either enhanced mixing or a lower lake level reducing the extent of Zone 3, which is compensated by an increase in the relative proportion of Zone 2 where Group I.1a

Thaumarchaeota are produced, thus lowering the BIT index (see section 2.2). The opposing trends in the records of these two proxies is clearly apparent in the last c. 160 kyr (R = -0.61, p < 0.001), and imply that both the BIT index and $IR_{6Me}$ register changes in lake hydrology through its influence on the relative size of the different mixing zones. Although these two moisture balance proxies are highly comparable, they also display important differences, such as the sharpness of recorded transitions in water-column structure. For example, during the period c. 113–99 ka (i.e., the Stage III lowstand), the BIT index displays

sustained low values, suggesting that the strong reduction in lake depth and enhanced upper water column mixing started and ended abruptly (Fig. 5a). The $IR_{6Me}$ during this period begins at relatively low values but steadily increases, indicating





a similarly dramatic, but more gradual decrease in lake depth (Fig. 5e). As the BIT index is controlled by the proliferation of Thaumarchaeota, which appear highly sensitive to changes in the upper mixed layer (i.e., they either bloom or fail to bloom depending on ambient conditions), BIT-index shifts through time can be expected to be more abrupt (Baxter et al., 2021). By contrast, the brGDGTs on which the $IR_{6Me}$ is based are produced in the anoxic zone of the lake, which makes their abundance less sensitive to environmental changes impacting the upper water column and thus generating a more paced response to gradual expansion/shrinking of the anoxic zones during phases of lake level rise or decline. The BIT index may thus be more sensitive and/or respond quicker to changes in monsoon strength than the $IR_{6Me}$ (as attested by its truthful registration of relatively modest drought events in the last 200 years: Buckles et al. 2016), although overall lake depth is likely the most important factor controlling the BIT index on long time scales. Events when both the BIT index and $IR_{6Me}$ infer abrupt changes in lake depth, for example, at c. 140 and 80 ka, thus likely do represent drastic changes in regional moisture balance within a relatively brief period of only a few hundreds of years. Also, the BIT index has a sensitivity threshold defined by the near absence of Thaumarchaeota (BIT index values approaching 1) under extreme shallow oxycline conditions, beyond which further changes to inferred wetter climatic conditions are no longer registered (Baxter et al., 2021). In the DeepCHALLA time series the BIT index is near 1 for a sustained period dated to c. 80–24 ka, largely overlapping with seismic Stage IV and suggesting persistent high lake level and extremely stratified water column conditions. The $IR_{6Me}$ varies continuously in this section, and therefore provides valuable additional information on lake-level or water-column changes during such intervals. Notably, the relationship between the relative proportion of 5Me- and 6Me-brGDGTs and lake depth documented in Lake Chala is not universal across all lake systems. Even in the similarly meromictic Lake Lugano (Switzerland) markedly different brGDGT distributions with depth occur (Weber et al., 2018), meaning that comprehensive local water column profiling of GDGT distributions is necessary prior to interpretation of down-core BIT index or $IR_{6Me}$ records.

### 5.3 Reliability of GDGT-based paleotemperature proxies in Lake Chala

Multiple lines of evidence (see section 5.1) suggest that GDGT variability in the deepest, oldest portion of the DeepCHALLA sediment sequence is not predominantly controlled by past climate variation. Therefore, discussion of the GDGT-based paleotemperature proxies ($TEX_{86}$, $MBT'_{5Me}$, MST) is conservatively constrained to the last 180 kyr of the record.

#### 5.3.1 $TEX_{86}$-based temperature reconstruction

Previous studies showed that the $TEX_{86}$ index of Lake Chala sediments is highly sensitive to contributions from non-Thaumarchaeotal sources, leading to incorrect palaeotemperature reconstructions (Sinninghe Damsté et al., 2012a; Buckles et al., 2013; Baxter et al., 2021). To distinguish between trustworthy and erroneous measurements, isoGDGT-0/cren, f[CREN'], and %isoGDGT-2 were applied to detect $TEX_{86}$ values which are likely influenced by mixed isoGDGT sources and therefore do not accurately reflect past temperature (Sinninghe Damsté et al., 2012a). Specific to Lake Chala, the importance of such non-Thaumarchaeotal sources can be assessed using threshold values informed by GDGT distributions in multi-year settling particle data and SPM profiles (Baxter et al., 2021). Accordingly, sediment horizons in the 180–0 ka section of the DeepCHALLA sequence with either BIT-index > 0.8, isoGDGT-0/cren > 0.7, f[CREN'] > 0.04 or %isoGDGT-2 > 45% (407 out of 798, or 51% of the to-





tal) indicate that the isoGDGT pool is not largely derived from (Group I.1a) Thaumarchaeota and, hence, that the associated
TEX$_{86}$ temperature estimates should be rejected. In particular, the overwhelming majority of sediments deposited during seis-
mic Stage IV must be excluded, preventing reconstruction of past temperature over most of the period c. 85-20 ka (Fig. 8b).
With exception of a TEX$_{86}$ increase from ∼0.45 to 0.6 after 20 ka (translating to a warming of > 10 °C using the calibration
of Tierney et al. 2010a), TEX$_{86}$ values are relatively stable at ∼0.45–0.55 throughout the last c. 180 kyr. This would imply a

comparable temperature regime during glacial and interglacial periods, which can be considered unrealistic as global climate
records indicate that large temperature variation occurred over these intervals, also at low latitudes (Blome et al., 2012). Al-
though the discussed thresholds offer some guidelines for omitting untrustworthy values, the need to exclude over half of the
sediment horizons and apparent lack of response to global temperature trends compromises the reliability of the TEX$_{86}$ index
as paleothermometer in Lake Chala.

### 5.3.2 BrGDGT-based temperature reconstruction

Variation in the MBT$'_{5Me}$ index throughout the 180-0 ka section of the DeepCHALLA sequence infers a temporal pattern
of temperature change which is likewise incompatible with global temperature variations reflected in the marine benthic
foraminiferal oxygen isotope stack (Lisiecki and Raymo, 2005) or with variations in atmospheric CO$_2$ concentration as prin-
cipal greenhouse-gas forcing (Petit et al., 1999). For example, although MBT$'_{5Me}$ correctly infers regional temperature to have

been higher than average during the last interglacial period (MIS5), the pattern of alternating warmer and cooler isotope sub-
stages MIS5a–MIS5e, with highest temperatures recorded during MIS5e, is not clear. Moreover, the inferred coldest interval
during the last glacial cycle is inferred to have occurred around 50 ka near the MIS3-MIS2 transition, rather than around the
Last Glacial Maximum (MIS2; Baxter et al. 2023). The abrupt increase and decrease in MBT$'_{5Me}$ at respectively c. 140 ka and
c. 80 ka occur simultaneously with major hydrological changes in Lake Chala as recorded by the BIT index and IR$_{6Me}$ (Figs. 5

and 8). This suggests that the imprint of climate-driven hydrological changes in Lake Chala on MBT$'_{5Me}$ is strong, and reason
to disqualify its use for temperature reconstruction at this site. Periodicity analysis of MBT$'_{5Me}$ over the last 180 kyr does
reveal periodicities of 24.8 kyr and 37.6 kyr, which may be linked to variation in solar insolation due to orbital precession and
obliquity, although only the obliquity-related periodicity can be considered significant using the 95% confidence interval. As
similar periodicities occur in the hydrological proxies (i.e., concentration of crenarchaeol, BIT index and IR$_{6Me}$; Fig. 9, Figs.

S9-S10), and as especially precession has a strong influence on tropical monsoon dynamics, this does not suffice to support its
applicability as temperature proxy for Lake Chala.

Recent paleoclimate studies in African lakes (e.g., Bittner et al. 2022), including Lake Chala (Baxter et al., 2023) indicate
that (some) 6-Me brGDGTs should be included in the calibration of brGDGT-based temperature proxies in order to achieve
more consistent temperature reconstructions. Application in DeepCHALLA of an earlier global lake calibration, which esti-

mates MST based on the combined abundance of 5-Me and 6-Me brGDGTs (Pearson et al., 2011), results in a temperature
reconstruction that displays peak temperatures of 25 °C reached during the current interglacial period (the last 11.7 ka) and
between c. 140 and c. 130 ka, likely (i.e., taking into consideration the chronological uncertainty) representing the globally
warmest episode of the last interglacial period (MIS5e) (Fig. 8e). Conversely, cool episodes centered at c. 150 ka, c. 60 ka and





15 ka correspond to known periods of extreme glaciation during the penultimate (MIS6) and last glacial periods (MIS4 and MIS2). Importantly, the MST time series does not include large shifts at c. 140 and c. 80 ka in conjunction with those of the BIT index and $IR_{6Me}$, suggesting that this temperature proxy is not affected by reorganization of water column structure related to changes in lake depth, and thus likely is a more trustworthy tracer of changes in regional air temperature. The MST record also does not display clear changes coincident with the sequence of low BIT-index and isoGDGT-0/cren values during the severe Stage III lowstand, further supporting the notion that the GDGT drivers of the MST proxy function independently from lake mixing. Therefore, it appears fair to suggest that the MST time series covering the last 180 ka of the DeepCHALLA sequence may constitute a reliable record of past regional temperature variation covering almost two glacial-interglacial cycles. This suggestion appears to be supported by clear periodicities of 23.0 kyr and 40.1 kyr in this data over the last 160 kyr, presumably reflecting the influences of orbital precession and obliquity on tropical African paleoclimate (Fig. S11e).

## 6   Conclusions

Analysis of iso- and brGDGTs in the 250-kyr sediment sequence from Lake Chala shows that the first c. 70 kyr of sedimentation are characterized by relatively low GDGT concentrations, and erratic variation in the BIT index and isoGDGT-0/cren ratio, suggesting a highly unstable oxycline position. Comparison with independent measures of lake-basin evolution, lake mixing and chemistry indicates that the structure of the water column during this early period was dissimilar to the present-day situation, because the lake had a leaky hydrology which prevented the accumulation of dissolved solids, thereby hampering chemical stratification and the formation of distinct mixing zones. Hence the differentiated niches of various GDGT producers as occurring in the water column today were not yet established, resulting in GDGT proxy values that are not predominantly controlled by climate variability. In line with this, time series of crenarchaeol concentration, the BIT index and $IR_{6Me}$ only start to display periodicities reflecting orbital insolaion forcing of the local climate from c. 180 ka onwards, suggesting that fromaround this time climate rather than lake basin evolution exerted the primary control on niches of GDGT producers and hence GDGT-derived proxies in the lake. The connection between GDGT proxies and regional climate as understood on the basis of modern-system studies gradually solidified between c. 180 ka and c. 80 ka as the lake developed the strong chemical gradient characterising modern-day conditions, permitting increasingly more trustworthy quantitative inferences of past climate regimes during those more recent periods.

The $IR_{6Me}$, which captures the relative proportion of 6Me-brGDGTs and 5Me-brGDGTs, is in Lake Chala related to past changes in lake depth, as this alters the relative size of the distinct niches were these lipids are most abundantly produced. Hence, it could be an important method for investigating past changes in regional moisture balance changes in this system, alongside the BIT index. Detailed consideration of available GDGT-based paleothermometers resulted in rejection of the $TEX_{86}$ temperature proxy, as previously set filtering criteria (Baxter et al., 2021) indicate that 51% of the sediment horizons younger than c. 180 ka likely contain large contributions of non-Thaumarchaeotal isoGDGTs. In particular, the strong influence of upper water-column mixing on Thaumarchaeota niche space casts doubt on the application of this temperature proxy in Lake Chala, and likely also other (tropical) lakes experiencing shallow oxycline conditions. The $MBT'_{5Me}$ index,





which is commonly assumed to best capture the temperature dependence of brGDGTs (Russell et al., 2018; Martínez-Sosa et al., 2021), results in a reconstruction that lacks a clear glacial-interglacial pattern and shows evidence for an overprint of lake mixing influences. Following research that shows the importance of including 6-Me brGDGTs in temperature proxies applied to Lake Chala (Baxter et al., 2023), MST reconstruction using the global lake calibration of Pearson et al. (2011) is found to display a strong and temporally feasible alternation between glacial and interglacial periods and major stadials, and contains clear periodicities related to the long-term variation in solar insolation due to orbital precession and obliquity.

Importantly, the types of chemical and physical changes that characterize the lake-system evolution of Lake Chala are not altogether unique, and similar processes are certainly involved in the history of most lake basins. To date these potential confounding factors are generally not considered when interpreting biomarker-based climate records from lakes. This work shows the necessity of applying a comprehensive approach which incorporates lake-basin information when interpreting down-core trends in sedimentary proxies to reconstruct past climate history, in particular when using biomarkers, like GDGTs, that are produced *in situ* in the water column or sediments. Based on our findings, particular caution is recommended when interpreting proxy records that extend to the initial filling stage of lakes or include episodes when lake-system functioning and sedimentation were clearly different from today, regardless of the apparent continuity of lacustrine deposition.

*Data availability.* Data from this chapter will be made available online upon publication.

*Author contributions.* Project administration was done by DV. Project conceptualization was done by DV, JSSD, FP and AJB. Funding aquisition, data curation, and resource procurement were done by DV and JSSD. FP, DV, JSSD and NW were responsible for supervision. Investigation was performed by AJB and AM. Formal analysis, visualization and writing of the original draft was done by AJB. All authors review and edited the

*Competing interests.* The author has declared that there are no competing interests.

*Acknowledgements.* This research was co-financed by NESSC Gravitation Grant 024.002.001 from the Dutch Ministry of Education, Culture and Science (OCW) to JSSD; by Ghent University Collaborative Research Operation grant BOF13/GOA/023, BRAIN-be project BR-121-A2 from the Belgian Science Policy Office (BelSPO), Hercules infrastructure grant AUGE/15/14-G0H2916N from the Research Foundation of Flanders, and a Francqui research professor mandate to DV; and by ICDP through the DeepCHALLA project (https://www.icdp-



online.org/projects/world/africa/lake-challa/). Recovery of the Lake Chala sediment record was facilitated by the government of Kenya through permit P/16/7890/10400 from the National Commission for Science, Technology and Innovation (NACOSTI), license EIA/PSL/3851 from the National Environmental Management Authority (NEMA), and research passes for foreign nationals issued by the Department of Immigration; and by the government of Tanzania through permits NA-2016-67 (270-285) and NA-2016-201 (277-292) from the Tanzania Commission for Science and Technology (COSTECH), permit EIA/10/0143/V.I/04 from the National Environmental Management Council, and resident permits issued by the Immigration Department. The lake-drilling operation was subject to environmental impact assessments conducted by Kamfor (Nairobi, Kenya) and Tansheq (Dar es Salaam, Tanzania), and permission from the Lands and Settlement Office of Taita-Taveta County (Kenya) to use government land as staging area.

The authors especially wish to thank our DeepCHALLA partners and the institutions of their affiliation in Kenya and Tanzania for project facilitation; Caxton Mukhwana Oluseno, the 'Air Force One' team, the Kamba and Taveta communities of Lake Chala area (villages of Challa, Kasokoni, Kidong and Nakuruto), and the DeepCHALLA team of field scientists for assistance in recovering the sediment record of Lake Chala; and the National Lacustrine Core Facility (LacCore) at the University of Minnesota (USA) for organizing the splitting, logging and initial processing of core samples. We are further grateful to F. Hilgen for assistance with periodicity analysis.



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
