# Peer review of "Disentangling influences of climate variability and lake-system evolution on climate proxies derived from isoprenoid and branched GDGTs: the 250-kyr Lake Chala record"

_EGUsphere, 2023_

## Author Response (AR1)

Dear Prof. De Jonge,

We thank you for reviewing our proposed changes and for the invitation to resubmit the manuscript titled, "Disentangling influences of climate variability and lake-system evolution on climate proxies derived from isoprenoid and branched GDGTs: the 250-kyr Lake Chala record" to Biogeosciences.

Both reviewers praise the extensiveness of our dataset, consisting of both branched and isoprenoid GDGT distributions for ~950 sediment horizons, and detailed interpretation of the time series of individual GDGTs and derived proxies. Reviewer #1 provides a few suggestions for further clarification of our text, whereas the main concern of reviewer #2 is the length of our manuscript, and suggests moving (parts of) our section summarizing previous work on Lake Chala (i.e., section 2) to supplementary files.

Besides incorporating the textual suggestions made by the reviewers, we have made the following changes to the new version of our manuscript. All our changes are marked in a "track changes" version of the complete manuscript file, with blue underlined text indicating new material and red strikethrough text indicating deleted material. Note that relocated text is present twice: in red (initial location) and blue (revised location).

- We have improved section 5.1 which describes the development of the chemical gradient in the lake over time by adding more explicit description of i) the evolution in lake hydrology over time due to gradual infilling of the crater basin with sediments and ii) the expected effects of major lake-level changes in the water column's dissolved ion content.
- We have made a clear effort to shorten (parts of) the manuscript, as requested by reviewer #2 and also suggested by the editor. Specifically, we have removed or condensed several parts of sections 1 (**Introduction**) and 2 (which now has the better fitting title '**The modern system and history of Lake Chala**'). This has led to a total reduction of ~120 lines from the main text (~1700 words; ~11% reduction). This condensing effort includes the section summarizing previous work (now an undivided subsection 2.5 with new title '**GDGT biochemistry**'). Therefore, we opted not to move that section to supplementary material for the following reasons: i) We feel that a (shortened!) overview of previous research is essential to understanding our interpretation of the data. Moving this part to the supplements would add references to this material in the discussion, which would alter the current flow of the text and likely lengthen the discussion. Removing this part would in turn require the reader of our work to first read a set of previous publications from different disciplines in order to be able to follow our interpretation of the GDGT data presented here. Hence, keeping the overview of previous work as part of our manuscript ensures that it can be read as a stand-alone publication by all disciplines.
ii) This paper is the first publication on the full ICDP-DeepCHALLA core. Hence, we feel that this is the appropriate and necessary moment to summarize the relevant research from this lake, providing the evolutionary and environmental context for all

forthcoming publications on this core. In addition, the fact that our system-level study deals with both geological and biological processes affecting organic geochemical climate proxies calls for an interdisciplinary journal with a wide audience, like Biogeosciences.

iii) Finally, we deliberately targeted Biogeosciences for publication of our work because it is open access, but also an electronic journal where publications only appear online. As such, publication should not be limited by the length of a manuscript as is the case in paper issues. In fact, moving sections from the main text to a supplement does not result in a net text reduction.

We hope that you find our revised version suitable for publication in Biogeosciences.

Sincerely,

Dr. Allix Baxter,

On behalf of all coauthors

Detailed list of changes made in response to the reviewers:

***Anonymous Referee #1:***
*I find a few places in the discussion where it seems that the authors are speculating. For example, is there other evidence for the development of the strong chemical gradient over time? This is certainly possible, but I didn't see any solid evidence for that. I don't think it is a problem to speculate in this type of paper, but they should be clear when they are doing so.* **We have clarified the explanation on the development of the chemical gradient in section 5.1.**

*The authors used the Pearson 2011 calibration to estimate temperature from brGDGTs, ultimately concluding that this "MST" reconstruction that combines the 5 and 6 Me GDGTs appears to be more appropriate to use in Lake Chala than MBT'5ME. This is somewhat surprising as there is substantial evidence that separating the two groups improves temperature reconstruction, though the logic they use is clearly laid out. I am very curious to see if either the Zhao et al 2023 (QSR), O'Beirne et al 2023 (GCA), or Wang et al 2024 (EPSL) calibrations provide more reasonable results. I fully understand that these calibrations were not yet published when this work was being performed, but now that they are out there I wonder if they do a better job that the Pearson calibration, which has issues of its own that compromise its application.*
**As motivated in our earlier replies to the reviewer we have chosen to not incorporate and discuss temperature reconstructions based on alternative calibrations into the present manuscript. In Baxter et al. (2023, Nature), we have already shown that any calibration based on the 'classically' defined MBT'5ME will result in reconstructed temperature records that are likely incorrect, and demonstrated that 6-me brGDGTs are required to better capture past temperature change in the Lake Chala sediment sequence. However, we do plan to discuss the potential of the new calibrations mentioned by the reviewer in a forthcoming publication in preparation which focusses primarily on the Lake Chala temperature record and its climatic implications.**

*A few minor typos:*
> *Line 164 needs closing parenthesis after johnson et al., 2016*
> *Line 351: change to "...Similar to brGDGTs, isoGDGT-0 is also produced"*
> *Lines 361 and 372: change to "Table 1"*
> *Line 461: Change to "...The degree of cyclisation (DC) of brGDGT was also calculated..."*
> *Line 497: change to "...within..."*
> *Lines 652-653: sentence is unclear, use of "respectively" twice in once sentence*
> *Line 676: change to "confirms"*
> *Line 722: change to "Stage"*
> *Line 796: change to "from"*
> *Line 886-887: used "inferred" twice*
> *Line 923: change to "insolation"*
> *Line 924: change to "from around"*

**We have made these changes.**

*You use both "GDGT" and "GDGTs" when referring to them as plural. Choose one and use it consistently*

**We have now consistently used "GDGTs" for the plural form in the revised manuscript.**

*Anonymous Referee #2*

*This paper incorporates a wealth of data and very detailed data interpretation. Finishing such a paper would be a tough task. The paper is generally well-written. However, reading a paper like this would be a challenge to a reader because it is too long and there is too much information that needs to be considered. I do not have comments on the scientific issues. Instead, I strongly suggest the authors remove some unnecessary parts of the paper (moved to the supplementary material) and make the paper succinct and easy to read. That would attract a much wider interest to read through the paper.*

**We have made a considerable effort to shorten the manuscript. Specifically, we have reduced the introduction and the section on the study site and previous work (section 2). As a result of this, the latter section is now entitled 'The modern system and history of Lake Chala', and summarized all previous biomarker work in one subsection rather than 3. In addition, the materials & methods section is condensed from 5 to 3 subsections. Overall, our efforts have led to a shortening of the text by ~120 lines and should have contributed to the overall readability of our manuscript.**

*Table 1 '%GDGT-2' A bracket is missing. Two 'iso-GDGT-2' were in the denominator.*
**We have corrected this.**

*L110 'only few' is not accurate. There are many downcore applications of brGDGTs in lakes.*
**We have changed this sentence as follows: "However, despite strong correlation between MBT′5Me in lacustrine surface sediments and temperature (Russell et al., 2018; Martínez-Sosa et al., 2021; Raberg et al., 2021), only few down-core applications of lake-based temperature calibrations have proved successful (Feakins et al., 2019; Stockhecke et al., 2021; Zhao et al., 2021; Zhang et al., 2021; Garelick et al., 2021; Ramos-Roman et al., 2022; Parish et al., 2023), partly due to continued uncertainty about the exact source(s) of brGDGTs in lakes."**

*L175 I suggest that you delete the detailed descriptions of all previous results or make a summary of them. Such a detailed description of previous GDGT work in Lake Chala makes the part look like a review. The paper is too long, which eliminates the interest of careful reading. This detailed description can be moved to the supplementary files*

**As motivated in our general response to the editor, we have opted not to move any parts of our manuscript to the supplements. We did, however, made a considerable effort to shorten the text, as specified in our reply to the first comment.**

*L260 delete 'then'*
*L439 following the method of Hopmans et al.(2016)*
*L459 calculated according to De Jonge et al. (2015).*
*Figure 10  in e) Stage VI should be stage IV*
*L796 from*
*L924 from around*
*L932 where these lipids*
*L951 Data on this chapter?  Which chapter?*
*L955 'review and edited the?'*
*L1033 Chen et al…. This is a preprint. The paper has been published in GCA.*

**We have corrected all these minor changes.**

---

## Author Response (AR2)

**Associate editor decision: Publish subject to minor revisions (review by editor)**
by Cindy De Jonge

**Public justification (visible to the public if the article is accepted and published):**
**Dear Allix Baxter and co-authors, thank you for responding to the comments of the co-authors in a timely fashion, and for accepting the suggestion to shorten and streamline the manuscript. Following this major revision I have read through the resubmitted version, and only have a few small editorial comments, as well as a suggestion to make the relevance of this study for globally distributed lake systems even more clear. I invite the authors to correct these small typo's and address my suggestion below.**

*We kindly thank the editor for their work on reviewing the resubmitted manuscript and providing further suggestions for improvement.*

L 132: "subapine zone" seems to be a typo
L 144, check punctuation around the Sepulchre reference.
L 193, check punctuation around Moernaut reference.
L 479, Table number is missing
L 492: %isoGDGT2: the ecological relevance of this ratio is not introduced.
L 564: Rewrite as: 'niches of GDGT producers'.
L 784: check punctuation around Baxter et al. reference.

*We have incorporated all minor changes suggested above.*

**Suggestion for making global relevance more clear:**
**Throughout the discussion there is not many comparisons made with other lakes. Is this because similar water column and/or sediment studies in the framework of the lake system evolution are uncommon? Or because Challa behaves differently than other lakes?**
*Certainly, studies of lake sediment records that both discuss the effects of basin evolution on GDGT climate proxies and are supported by adequate water-column investigation are still very uncommon. The limited water column studies that are currently available indicate that oxycline depth is a major controlling factor on the abundance and spatio-temporal distribution of GDGTs in stratifying lakes. However, how exactly GDGTs respond to changes in oxygen availability may differ between lakes globally, as we have addressed in our earlier work (e.g., van Bree et al., 2020 BGS on brGDGTs and Baxter et al., 2021 QSR on isoGDGTs) and has also been addressed elsewhere (e.g., Sinninghe Damsté et al., 2022, QSR). This is also the reason why we have not advocated here or previously (Baxter et al., 2021, QSR; Baxter et al., 2023, Nature) for applying GDGT proxies validated for use in Lake Chala (e.g., the BIT index) to other stratifying lakes without first conducting thorough modern-system studies. Instead, we argue that it is key to establish whether lake sedimentary archives reflect stable water column conditions throughout lake history prior to downcore application and interpretation of GDGT proxies as climate records.*

*We emphasized this issue in the final sentences of section 5.2 of the discussion in relation to the 5- and 6-me brGDGT distribution in Lake Chala with the following text (updated slightly from previous version):*

*"Certainly, the exact relationship between 5Me- and 6Me-brGDGTs and the particular water column zones which allows their relative proportion to be used as a qualitative proxy for lake depth in Lake Chala is not likely universal across different lake systems. Even in the similarly meromictic Lake Lugano (Switzerland) markedly different brGDGT distributions with depth occur (Weber et al., 2018) implying that comprehensive local water-column profiling of GDGT distributions is necessary prior to interpretation of down-core BIT index or IR$_{6me}$ records".*

**To address the 'uniqueness' of Lake Challa, could the authors include a statement in their conclusion whether the interpretations on GDGT niches and derived temperature ratios based on low latitude meromictic Lake Challa are fully/in part/not at all applicable to shallow (seasonally stratified or fully mixed) lakes or lakes at higher latitudes (seasonally variable temperature and a colder hypolimnion)? Would a shallow lake be comparable with a Lake Challa Zone 1-3 for instance.**

*We refer the editor to sections 4.5 and 4.6 of Baxter et al., 2021, QSR where we included a comparison of the isoGDGT SPM data set to other studies of isoGDGTs in diverse lake systems and additionally speculated about the functioning of our presented upper water-column stratification proxies (e.g., BIT, f[CREN'], isoGDGT-0/cren) in lakes shallower and deeper than Lake Chala. We have added the following statement to our conclusion in order to stress that our results are likely not applicable to lakes in other regions / mixing regimes:*

*"We stress that the niche partitioning of GDGT producers within the distinct water-column mixing zones of Lake Chala underpinning the temperature proxies as presented here is potentially unique (as discussed previously in Baxter et al., 2021, QSR). Dimictic and monomictic lakes, as are common in cold- and warm-temperate regions, may be equivalent to zones 1-2 or zones 1-3 of Lake Chala depending on their trophic status (i.e., level of primary productivity), as this would control the occurrence and seasonal persistence of hypolimnetic anoxia. Unstratifying (truly shallow) lakes enjoy year-round oxygen injection and nutrient upcycling, and thus present GDGT producers with entirely different niches. At the other extreme of the depth gradient, permanent stratification of the lower water column (meromixis) can be created and maintained by different processes, and with different prevalence in temperate and tropical lakes (Gulati et al. 2017), consequently the occurrence of GDGT niches as in zones 4-6 of Lake Chala is more likely site-specific (and as documented here, may change over long time scales). Thorough modern-system studies should ideally be conducted before applying such proxies to the sedimentary record of any type of lake system.".*

*Gulati, R. D., Zadereev, E. S., & Degermendzhi, A. G. (Eds.). (2017). Ecology of meromictic lakes (Vol. 228). Cham: Springer.*

**Furthermore, the authors might want to 'philosophize' how we interpret the fact that the reconstructed sedimentary temperatures (based on brGDGTs) rely on the contribution of two different water layers and potentially different communities**

**of producers (zone 3 vs zone 5 and 6)? How does this for instance relate back to the mechanism that a single brGDGT producer modulates the lipid composition of their membrane to regulate the membrane fluidity?**

*It is possible that 5- and 6-Me brGDGTs are produced by different communities, as we have discussed in van Bree et al. (2020, BGS), and that changes in community structure may affect the temperature signal recorded by the MST proxy. Work on Lake Lugano by Weber et al (2018, PNAS) likewise suggested that brGDGTs likely have distinct bacterial producers in different parts of the water column. However, given the fundamental work showing that methylation in a wide range of membrane lipids is adjusted according to temperature (e.g. Suutari & Laakse, 1994; Chen et al., 2022) and the fact that studying only the core lipids (e.g., without the polar head groups) misses critical information about the full structure of the original membrane, we do not believe it is appropriate to assume that community structure, and not membrane adaptation is the prime driver of these relationships. Considering this, and that we have already speculated about the producers of brGDGTs and the consequences for temperature proxies in van Bree et al. (2020, QSR) we chose to not discuss this further in the present manuscript which focusses on assessing the application of paleotemperature proxies to the Lake Chala sediments.*

*Chen, Y., Zheng, F., Yang, H., Yang, W., Wu, R., Liu, X., ... & Zeng, Z. (2022). The production of diverse brGDGTs by an Acidobacterium providing a physiological basis for paleoclimate proxies. Geochimica et Cosmochimica Acta, 337, 155-165.*

*Suutari, M., & Laakso, S. (1994). Microbial fatty acids and thermal adaptation. Critical reviews in microbiology, 20(4), 285-328.*

*We hope with these minor changes our manuscript is now ready for publication in Biogeosciences.*

*Sincerely,*
*Dr. Allix Baxter*
*On Behalf of all co-authors*